

# Reducing Temporal Uncertainty in Soil Bulk Density Estimation Using Remote Sensing and Machine Learning Approaches

Sunantha Ousaha[12], Zhenfeng Shao[1], Zeeshan Afzal[1]

[1]State Key Laboratory of Information Engineering in Surveying, Mapping and Remote Sensing, Wuhan University, Wuhan,

430072, China

[2]Soil Resources Survey and Research Division, Land Development Department, Ministry of Agriculture and Cooperatives,

Bangkok, 10900, Thailand

*Correspondence to*: Sunantha Ousaha (Sunantha.sun@hotmail.com)

**Abstract.** Soil bulk density (BD), a key physical property affecting soil compaction, porosity, and carbon stock estimation,

exhibits considerable spatial and temporal variability. However, current BD estimation methods especially traditional

pedotransfer functions (PTFs) are inherently static and not designed for temporal analysis. This presents a significant limitation

for soil monitoring across large and heterogeneous regions. In this study, we developed a machine learning (ML) approach

integrated with remote sensing data to map and monitor BD across Thailand from 2004 to 2009 at national scale. We used

multispectral indices, topographic variables, climate data, and organic carbon content to train six ML models: Artificial Neural

Networks (ANN), Deep Neural Networks, Random Forest, Support Vector Regression, XGBoost, and LightGBM. Model

performance was evaluated using in-situ BD measurements from 236 soil samples collected in 2004. For benchmarking

purposes, 76 published PTFs were also assessed on the same dataset. Results showed that the ANN model achieved the highest

prediction accuracy ($R^2 = 0.986$; RMSE = 0.017 g cm$^{-3}$), outperforming both other ML models and all PTFs. Temporal analysis

using the ANN model revealed a 7.27% increase in mean BD and a 41.23% reduction in standard deviation between 2004 and

2009, indicating increased soil compaction and reduced variability. Feature importance analysis identified organic carbon,

vegetation indices, slope, and temperature as the most influential variables. The resulting high-resolution BD maps captured

national-scale spatial and temporal trends and provide a robust foundation for soil quality monitoring, carbon accounting, and

sustainable land use planning in tropical agroecosystems.

## 1 Introduction

Soil bulk density (BD) is a critical parameter in soil science, playing a pivotal role in the measurement of soil organic carbon

(SOC) stocks and influencing a wide range of soil properties (Demir et al., 2022). Surface BD is dominated factor controlling

soil porosity and compaction (Li et al., 2019), which in turn impact root penetration, water movement, and microbial activity.

Despite its importance, obtaining reliable BD measurements remains a challenge. Traditional methods, such as core sampling,



are labor-intensive, time-consuming, and spatially limited, especially in large and heterogeneous landscapes. Consequently,
indirect approaches like PTFs and advanced techniques incorporating remote sensing and machine learning have emerged as
promising alternatives for large-scale BD estimation.

Pedotransfer Functions (PTFs) have long been used to estimate BD by predicting soil properties based on readily available
soil attributes. PTFs are widely utilized to estimate SOC stocks across different scales (Manuel Rodríguez-Rastrero, 2022).
Early work by Manrique and Jones (1991) established one of the first PTFs to estimate BD using organic carbon. PTF models
have incorporated additional variables, including fine earth fractions, organic carbon (OC), organic matter (OM), and particle
size fractions (Patil and Singh, 2016). Schillaci et al. (2021) explored using soil and environmental data for PTFs, achieving
variable but promising results. While PTFs offer a cost-effective and accessible approach, they are inherently limited by the
mathematical assumptions of their underlying models and the quality of the data used to derive them (Vasiliniuc and Patriche,
2015). This makes PTFs less effective in predicting BD across heterogeneous landscapes, as noted by Nasta et al. (2020) and
Sevastas et al. (2018). Therefore, the integration of diverse data sources and advanced modelling techniques is crucial to
overcome these limitations.

The advent of remote sensing technologies has significantly enhanced the potential for BD estimation. Multispectral,
hyperspectral, LiDAR, and synthetic aperture radar (SAR) sensors, deployed on satellite and airborne platforms, provide
extensive spatial coverage and enable detailed soil property mapping (Poggio and Gimona, 2017). These sensors, including
Landsat 7, Landsat 8, Sentinel-1/2, Hyperion, and various spectroscopy techniques, offer new possibilities for improving BD
prediction accuracy (Yang and Guo, 2019). Integrating multispectral and SAR data has proven particularly effective, as it
captures complementary information on soil surface properties and structure (Hengl et al., 2017). Hyperspectral imagery and
LiDAR-derived covariates have also shown strong correlations with soil properties and spatial distribution patterns (Guo et
al., 2021; Pittman and Hu, 2020). Soil characteristics influenced by plant cover have shown strong correlations with
hyperspectral spectra (Anne et al., 2014). In contrast, vis–NIR spectra from spectroscopy did not show significant differences
in performance compared to PTFs-based models, but were still superior (Katuwal et al., 2020). However, while remote sensing
data significantly contribute to BD prediction, studies suggest that combining these inputs with machine learning (ML) models
can further enhance accuracy and efficiency

Recent machine learning (ML) algorithms outperform regression functions and geostatistical methods in BD prediction due to
their ability to capture non-linear relationships and complex interactions among soil properties (Anne et al., 2014; Panagos et
al., 2024). Traditional methods like regression functions and geostatistical techniques such as Kriging (Poggio and Gimona,
2017) and Variograms are often limited by their linear assumptions and inability to manage complex soil variable interactions
(Padarian et al., 2020), leading to issues such as overestimation (Panagos et al., 2024). In contrast, ML algorithms excel in
these areas by leveraging large datasets to identify intricate patterns and relationships. Studies have demonstrated that advanced
ML models, such as random forest (Hengl et al., 2017), artificial neural networks (Aitkenhead and Coull, 2020), support vector
Machines (Hateffard et al., 2023), and extreme gradient boosting (Salehi Hikouei et al., 2021) offer significant improvements
in prediction accuracy. These models can efficiently account for spatial variability and autocorrelation, providing more fast,



reliable (Kim et al., 2023), and precise BD predictions across heterogeneous landscapes (Guo et al., 2021). However, ML models have limitations which can be challenging to acquire and manage (James et al., 2013). ML models often act as "black boxes," making their predictions difficult to interpret and validate (Rudin, 2019). Models trained on specific regional data may not generalize well to other areas, necessitating periodic retraining with updated data to maintain accuracy (Panagos et al., 2024). Despite advances in BD prediction using remote sensing and machine learning, most studies focus on spatial accuracy, with limited attention to year-to-year variability and temporal uncertainty at national scales.

In this study, we develop a machine learning framework that integrates remote sensing, environmental variables, and in-situ organic carbon data to estimate soil bulk density (BD) at national scale. We train six machine learning models, Artificial Neural Networks (ANN), Deep Neural Networks (DNN), Random Forest (RF), Support Vector Regression (SVR), XGBoost, and LightGBM using soil and spectral data from 2004. The best-performing model is then applied to 2009 satellite and OC data to predict BD and analyze temporal changes. To benchmark performance, we evaluate 76 published pedotransfer functions (PTFs) using the same 2004 dataset. Our objectives are to (a) identify the most accurate and scalable ML approach for BD prediction using satellite-derived covariates, (b) quantify temporal changes and uncertainty in BD between 2004 and 2009, and (c) compare ML-based predictions against conventional PTFs. These findings enable accurate, scalable, and temporally responsive monitoring of soil bulk density for improved land management and resource planning.

## 2 Materials and methods

### 2.1 Soil Data Collection

Historical soil samples were collected by the Land Development Department (LDD), Ministry of Agriculture and Cooperatives in Thailand, were utilized in this study using random sampling strategy that cover soil and crop type in agriculture area. The dataset was prepared to explore and develop a robust ML model. Figure 1 illustrates a dataset of 236 soil samples collected in 2004, which were used for model development. Additionally, soil samples with OC data collected in 2009 were used for model implementation. These samples included measurements of soil bulk density (g cm⁻³), organic carbon (OC) content, organic matter (OM), and particle size (sand, silt, clay). BD was collected by undisturbed soil core sampling method by a cylindrical core sampler for bulk density analysis (Fao, 2023; Holliday, 1990) based on the following equation (1). Additionally, Walkley and Black's method was used to analyse the soil sample pits to determine the percentage of OC calculated following Eq. (1),

$$\text{Bulk density (BD)} = \frac{(\text{dry soil mass} + \text{cylinder mass}) - \text{cylinder}}{\text{volumn of the cylinder}} \times 100, \tag{1}$$

$$\text{BD} = \frac{\text{dry soil mass (g)}}{\text{volume of cylinder (cm}^3)} \times 100, \tag{2}$$

The volume of the cylinder is defined in Eq. (3),

$$V_s = \pi r^2 h, \tag{3}$$



where $\pi$ is 3.1416, $r$ is radius (half of the diameter) (cm), and $h$ is height of the core (cm)

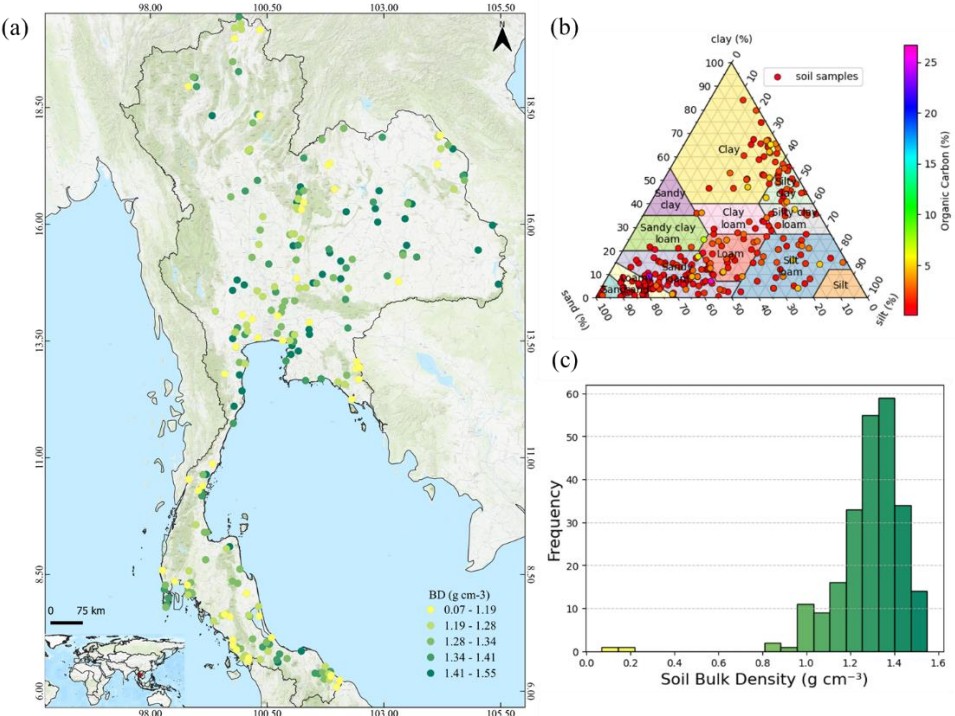

**Figure 1** Study area map and location: (a) spatial distribution of soil sampling points across the study area, (b) USDA soil texture triangle diagram, and (c) histogram depicting the frequency distribution of soil bulk density (g cm⁻³).

## 2.2 Remote Sensing Data Collection and Preprocessing

All remote sensing data were obtained from Earth Engine data catalog, consisting of Landsat 5 thematic mapper (TM) imagery, topographic data, and climate data.

### 2.2.1 Landsat 5 Thematic Mapper (TM) and Pre-processing

The study area experiences frequent cloud cover due to its temperate climate, posing a significant challenge for satellite image analysis. To address this, Landsat 5 Thematic Mapper (TM) imagery for the years 2004 and 2009 was processed to correct the entire year's dataset and ensure reliable input for subsequent analysis. The imagery consists of six spectral bands spanning the visible, near-infrared (NIR), and shortwave infrared (SWIR) regions, each with a 30-meter spatial resolution. Landsat 5 TM Level 2 images were obtained from the Google Earth Engine (GEE) data catalog, leveraging its extensive repository and advanced tools for cloud correction and quality enhancement.



The cloud masking process utilized the QA band to identify and remove cloud-affected pixels across the entire dataset. Relevant bits in the QA band, specifically bit 3 for cloud shadows and bit 5 for clouds, were used to generate a comprehensive
cloud mask. This ensured that only clear pixels were retained for analysis. To further mitigate the impact of residual clouds and anomalies, a weighted median composite image was created for each year, combining cloud-free pixels throughout the entire year. This composite approach minimized noise and temporal inconsistencies, providing a robust foundation for deriving key predictive indices. Vegetation indices, moisture indices, and soil indices were calculated from the composite images as covariates for BD estimation. These indices, tailored to capture the environmental variability of the study area, are detailed in
Table 1.

**Table 1.** List of commonly used vegetation indices, soil indices, and moisture index relating to soil bulk density (BD)

| Spectral Index | Abbreviate | Formular |
| --- | --- | --- |
| Normalized Difference Vegetation Index | NDVI | (Band 4 – Band 3) / (Band 4 + Band 3) |
| Enhanced Vegetation Index | EVI | 2.5 × ((Band 4 – Band 3) / (Band 4 + 6 × Band 3 – 7.5 × Band 1 + 1)) |
| Bare Soil Index | BSI | (Band 5 + Band 3) - (Band 4 + Band 1) / (Band 5 + Band 3) + (Band 4 + Band 1) |
| Modified Soil Adjusted Vegetation Index | MSAVI | (2 × Band 4 + 1 – sqrt ((2 × Band 4 + 1)$^2$ – 8 × (Band 4 – Band 3))) / 2 |
| Soil Adjusted Vegetation Index | SAVI | ((Band 4 – Band 3) / (Band 4 + Band 3 + 0.5)) × (1.5) |
| Clay Index | CI | (SWIR1 - SWIR2) / (SWIR1 + SWIR2) |
| Normalized Difference Soil Index | NDSoI | (Band 7 – Band 2) / (Band 7 + Band 2) |
| Dry Bareness Soil Index | DBSI | (SWIR1-Green/SWIR1+Green)-NDVI |
| Normalized Difference Moisture Index | NDMI | (Band 4 – Band 5) / (Band 4 + Band 5) |

### 2.2.2 Topography

Various terrain attributes were derived, including elevation, slope, and aspect, from the Shuttle radar topography mission SRTM Digital Elevation Data Version 4, which has an original pixel size of 90 meters. These data were used to obtain topographic data with a resampling at 30 m.

### 2.2.3 Climate data

For this study, TerraClimate data was utilized to obtain climate variables for the years 2004 and 2009. TerraClimate provides
gridded datasets of precipitation accumulation and mean temperature at global scales, synthesized from weather station data. The data used in this study represents mean values at a specific spatial resolution of 4638.3 meters meters.



## 2.3 Existing 76 Pedotransfer Functions (PTFs)

This study evaluates 76 published PTFs using in-situ soil data from 2004 to identify the best-performing model for soil bulk
density (BD) prediction, detailed in Table 2. The selection of these PTFs was guided by their reliance on accessible and
practical soil properties, including sand, silt, and clay contents, organic carbon (OC), organic matter (OM), and soil depth.
These properties are widely used due to their cost-effectiveness and ease of measurement. The PTFs are categorized into five
primary groups based on their input variables: 26 rely exclusively on OM, 21 utilize OC as the primary predictor, 4 are
dependent solely on particle size (PS) fractions, 6 combine particle size and OM (PSOM), and 18 integrate particle size and
OC (PSOC). The PTF with the lowest Root Mean Square Error (RMSE) will be selected and applied to 2009 soil data for BD
calculation, enabling a comparative analysis of temporal BD changes with values derived from the ANN model and remote
sensing data.

**Table 2.** List of the 76 published PTFs with references

| TPFs | Year | Function | Reference |
|---|---|---|---|
| OM1 | 1964 | $BD = 10^{[2.09963 - 0.00064(\text{logom}) - 0.22302(\text{logom})2]}/100$ | (Curtis and Post, 1964) |
| OM2 | 1966 | $BD = 1.53 - 0.05 \cdot om$ | (Saini, 1966) |
| OM3 | 1970 | $BD = 1.482 - 0.6786 \cdot \log_{10} om$ | (Jeffrey, 1970) |
| OM4 | 1973 | $BD = 100/((om/K_1) + (100 - om/K_2))$ | (Adams, 1973) |
| OM5 | 1973 | $BD = 1/(0.6268 + 0.0361 \cdot om)$ | (Drew, 1973) |
| OM6 | 1983 | $BD = \exp(-2.314 - 1.0788 \times \log(om / 100) - 0.1132 \times (\log(om / 100))^2)$ | (Federer, 1983) |
| OM7 | 1983 | $BD = \exp(-2.314 - 1.0788 \times \log(om) - 0.1132 \times (np.\log(om))^2)$ | (Federer, 1983) |
| OM8 | 1983 | $BD = 0.111 \cdot 1.45/(1.45 \cdot om /100 + 0.111 \cdot (1 - om /100))$ | (Federer, 1983) |
| OM9 | 1981 | $BD = 1.558 - 0.728 \cdot \log_{10} om$ | (Harrison and Bocock, 1981) |
| OM10 | 1982 | $BD = 100/((om /K_1) + (100 - om)/K_3)$ | (Rawls and Brakensiek, 1982) |
| OM11 | 1989 | $BD = 0.075 + 1.301 \cdot \exp(-0.06 \cdot om)$ | (Grigal et al., 1989) |
| OM12 | 1989 | $BD = \exp(-2.39 - 1.316 \cdot \ln(om /100) - 0.167 \cdot (\ln(om /100))^2)$ | (Huntington et al., 1989) |
| OM13 | 1989 | $BD = 1/(0.564 + 0.0556 \cdot om)$ | (Honeysett and Ratkowsky, 1989) |
| OM14 | 1989 | $\ln BD = -2.39 - 1.316 \cdot \ln(om) - 0.167(\ln(om))^2$ | (Huntington et al., 1989) |
| OM15 | 1994 | $BD = 1.565 - 0.2298 \cdot om^{0.5}$ | (Tamminen and Starr, 1994) |
| OM16 | 2000 | $BD = 100/[(om/0.244) + ((100 - om)/MBD_1)]$ | (Post and Kwon, 2000) |
| OM17 | 2002 | $BD = (0.12 \times 1.4)/(om/100 \cdot 1.4 + (1 - om/100) \times 0.12)$ | (Tremblay et al., 2002) |
| OM18 | 2004 | $BD = \exp(-1.81 - 0.892 \cdot \ln(om/100) - 0.092 \cdot \ln(om/100)^2)$ | (Prevost, 2004) |
| OM19 | 2004 | $BD = \exp(-2.314 - 1.0788 \cdot \ln(om/100) - 0.1132 \cdot \ln(om /100)^2)$ | (Dexter, 2004) |
| OM20 | 2005 | $BD = 1.775 - 0.173 \cdot om^{0.5}$ | (De Vos et al., 2005) |
| OM21 | 2006 | $BD = 100 / (om / 0.244 + (100 - om) / 1.41)$ | (Cienciala et al., 2006) |
| OM22 | 2008 | $BD = (0.111 \cdot 1.767)/(1.767 \cdot om/100 + [(1 - om/100) \cdot 0.111])$ | (Perie and Ouimet, 2008) |



| OM23 | 2008 | $BD = 100/(om/0.244 + (100-om)/1.41)$ | (Perie and Ouimet, 2008) |
|---|---|---|---|
| OM24 | 2012 | $BD = \exp(0.5379 - 0.0653 \cdot om^{0.5})$ | (Han et al., 2012) |
| OM25 | 2014 | $BD = 100/(OM/0.14 + (100-om)/1.153]$ | (Nanko et al., 2014) |
| OM26 | 2023 | $BD = 0.348 + 0.993 \times \exp(-0.0882 \times om)$ | (Tao et al., 2023) |
| OC1 | 1970 | $BD = 1.37 - 0.076 \cdot oc$ | (Williams, 1971) |
| OC2 | 1980 | $BD = 1.72 - 0.294 \times oc^{0.5}$ | (Alexander, 1980) |
| OC3 | 1991 | $BD = 1.510 - 0.113 \cdot oc$ | (Manrique and Jones, 1991) |
| OC4 | 1986 | $BD = 1.446 - 0.000645 \cdot depth - 0.344 \cdot \log_{10}(oc)$ | (Zinke et al., 1986) |
| OC5 | 1989 | $\ln BD = 0.263 - 0.147 \cdot \ln(oc) - 0.103(\ln(oc))^2$ | (Huntington et al., 1989) |
| OC6 | 2003 | $BD = 1.2901 - 0.1229 \cdot \ln(oc)$ | (Wu et al., 2003) |
| OC7 | 2005 | $BD = 1.608 - 0.0872 \cdot oc$ | (Valzano F et al., 2005) |
| OC8 | 2005 | $BD = 1.780 - 0.379 \cdot oc^{0.5} + 0.00123 \cdot depth$ | (Heuscher et al., 2005) |
| OC9 | 2005 | $BD = 1.3565 \cdot \exp(-0.0046 \cdot oc \cdot 10)$ | (Song et al., 2005) |
| OC10 | 2005 | $BD = 1.3770 \cdot \exp(-0.0048 \cdot oc \cdot 10)$ | (Song et al., 2005) |
| OC11 | 2007 | $BD = 0.29 + 1.2033 \cdot \exp(-0.075 \cdot oc)$ | (Yang et al., 2007) |
| OC12 | 2009 | $BD = 2.684 - 140.943 \times 0.006) \times \exp(-0.006 \cdot oc)$ | (Ruehlmann and Körschens, 2009) |
| OC13 | 2009 | $BD = (2.684 - 140.943 \cdot 0.008) \cdot \exp(-0.008 \cdot oc \cdot 10)$ | (Ruehlmann and Körschens, 2009) |
| OC14 | 2011 | $BD = 1.4842 - 0.1424 \cdot oc$ | (Kobal et al., 2011) |
| OC15 | 2012 | $BD = 1.4903 - 0.33293 \cdot \ln(oc)$ | (Hollis et al., 2012) |
| OC16 | 2015 | $BD = 0.701 + 0.952 \cdot \exp(-0.29 \cdot oc)$ | (Hossain et al., 2015) |
| OC17 | 2018 | $BD = 1.448 \exp^{(-0.03(oc)}$ | (Abdelbaki, 2018) |
| OC18 | 2016 | $BD = 1.705925 - 0.342497 \cdot oc^{0.5}$ | (Reidy et al., 2016) |
| OC19 | 2017 | $BD = 1.197 \times oc^{-0.229}$ | (Atwood et al., 2017) |
| OC20 | 2018 | $BD = 1/0.733 + 0.0982 \times (oc/100)$ | (Chen et al., 2018) |
| OC21 | 2018 | $BD = 2.039 - 0.563 \cdot oc + 0.103 \cdot oc^2$ | (Sevastas et al., 2018) |
| OC22 | 2024 | $BD = 0.4527 + 1.0816 \times \exp(-0.2155 \cdot oc)$ | (Do et al., 2024) |
| PS1 | 1998 | $BD = 1.352 - 0.0045(cl)$ | (Bernoux et al., 1998) |
| PS2 | 2007 | $BD = 1.5224 - 0.0005(cl)$ | (Benites et al., 2007) |
| PS3 | 2007 | $BD = 1.35 + 0.0045 \cdot sa + (44.7 - sa)^2 \cdot (-6 \cdot 10^{-5}) + 0.060 \cdot \ln(depth)$ | (Tranter et al., 2007) |
| PS4 | 2016 | $BD = 1.177 + 0.00263 \cdot sa - 0.0439 \cdot \ln(si) + 0.00208 \cdot si$ | (Akpa et al., 2016) |
| PSOM1 | 2004 | $x = -1.2141 + 4.23123 \cdot sa/100;\ y = -1.70126 + 7.55319 \cdot cl/100$ | (Rawls et al., 2004) |
| | | $z = -1.55601 + 0.507094 \cdot om;\ w = -0.0771892 + 0.256629 \cdot x +$ | |
| | | $0.256704 \cdot x^2 - 0.140911 \cdot x^3 - 0.0237361 \cdot y - 0.098737 \cdot x^2 \cdot y - 0.140381 \cdot y^2 +$ | |
| | | $0.0140902 \cdot x \cdot y^2 + 0.0287001 \cdot y^3$ | |
| | | $BD = 1.36411 + 0.185628 \cdot (0.0845397 +$ | |
| | | $0.701658 \cdot w - 0.614038 \cdot w^2 - 1.18871 \cdot w^3 + 0.0991862 \cdot y - 0.301816 \cdot w \cdot y - 0$ | |
| | | $.153337 \cdot w^2 \cdot y - 0.072242 \cdot y^2 + 0.392736 \cdot w \cdot y^2 +$ | |



$0.0886315 \cdot y^3 - 0.601301 \cdot z + 0.651673 \cdot w \cdot z - 1.37484 \cdot w^2 \cdot z + 0.298823 \cdot y \cdot z - 0.192686 \cdot w \cdot z \cdot y + 0.0815752 \cdot y^2 \cdot z - 0.0450214 \cdot z^2 - 0.179529 \cdot w \cdot z^2 - 0.0797412 \cdot y \cdot z^2 + 0.00942183 \cdot z^3)$

| | | | |
|---|---|---|---|
| PSOM2 | 1957 | $BD = 1.8014 - 0.8491 \cdot \log_{10}(om + 2) + 0.0026 \cdot cl$ | (Eschner et al., 1957) |
| PSOM3 | 2004 | $BD = 1/(0.59 + 0.00163 \cdot cl + 0.0253 \cdot om)$ | (Dexter, 2004) |
| PSOM4 | 2010 | $BD = 1.308 + 0.0119 \cdot cl + 0.0103 \cdot sa - 0.00018 \cdot cl^2 - 0.00008 \cdot sa^2$ $0.00062 \cdot si \cdot om - 0.00059 \cdot sa \cdot om$ | (Keller and Håkansson, 2010) |
| PSOM5 | 2011 | $BD = 100/(om/0.224 + (100-om)/(0.935 + 0.049 \cdot \log_{10}(depth) + 0.0055 \cdot sa + 0.000065 \cdot (sa-38.96)^2))$ | (Minasny and Hartemink, 2011) |
| PSOM6 | 2013 | $BD = 100/(om/0.224 + (100 - om)/[1.017 + 0.0032 \cdot sa + 0.054 \cdot \log_{10}(depth)])$ | (Hong et al., 2013) |
| PSOC1 | 1998 | $BD = 1.578 - 0.054 \times oc^{-0.006} \times si^{-0.004} \times cl$ | (Tomasella and Hodnett, 1998) |
| PSOC2 | 1998 | $BD = 1.398 - 0.0047 \cdot cl - 0.042 \cdot oc$ | (Bernoux et al., 1998) |
| PSOC3 | 1998 | $BD = 0.87 + 0.071 \cdot \ln(cl) + 0.093 \cdot \ln(sa) - 0.254 \cdot \ln(oc)$ | (Hallett et al., 1998) |
| PSOC4 | 1998 | $BD = 1.46 - 0.0254 \cdot \ln(cl) + 0.0279 \cdot \ln(sa) - 0.261 \cdot \ln(oc)$ | (Hallett et al., 1998) |
| PSOC5 | 2000 | $BD = 1.70398 - 0.00313 \cdot si + 0.00261 \cdot cl - 0.11245 \cdot oc$ | (Leonaviciute, 2000) |
| PSOC6 | 2001 | $BD = 1.673 - 0.0071 \cdot oc - 0.0017 \cdot si - 0.003 \cdot cl$ | (Calhoun et al., 2001) |
| PSOC7 | 2002 | $BD = \exp(0.313 - 0.191 \cdot oc + 0.02102 \cdot cl - 0.0004768 \cdot cl^2 - 0.00432 \cdot si)$ | (Kaur et al., 2002) |
| PSOC8 | 2005 | $BD = 1.711 - 0.0487 \cdot oc^2 + 0.0059 \cdot oc^3 + 0.002 \cdot cl$ | (Heuscher et al., 2005) |
| PSOC9 | 2005 | $BD = 1.674 - 0.310 \cdot oc^{0.5} + 0.015 \cdot cl - 2.41 \cdot 10^{-4} \cdot si^2$ | (Heuscher et al., 2005) |
| PSOC10 | 2008 | $BD = 1.5688 - 0.0005(cl) - 0.009(oc)$ | (Benites et al., 2007) |
| PSOC11 | 2009 | $BD = 1.386 - 0.078 \cdot oc + 0.001 \cdot si + 0.001 \cdot cl$ | (Men et al., 2008) |
| PSOC12 | 2012 | $BD = 0.80806 + 0.823844 \cdot \exp(-0.27993 \cdot oc) + 0.0014065 \cdot sa - 0.0010299 \cdot cl$ | (Hollis et al., 2012) |
| PSOC13 | 2012 | $BD = 0.69794 + 0.750636 \exp(-0.230550 \cdot oc) + 0.0008687 \cdot sa - 0.0005164 \cdot cl$ | (Hollis et al., 2012) |
| PSOC14 | 2013 | $BD = 1.228 - 0.155 \times \log(oc) + 0.008 \cdot sa$ | (Al-Qinna and Jaber, 2013) |
| PSOC15 | 2015 | $BD = 1.64581 - 0.00362(cl) - 0.0016 \cdot sa - 0.0158 \cdot oc$ | (Botula et al., 2015) |
| PSOC16 | 2017 | $BD = (1.6179 - 0.0180 \cdot (cl + 1)^{0.46} - 0.0398 \cdot oc^{0.55})^{.33}$ | (Beutler et al., 2017) |
| PSOC17 | 2018 | $BD = 2.268 - 0.179 \times \ln(sa) - 0.345 * \ln(oc)$ | (Sevastas et al., 2018) |
| PSOC18 | 2024 | $BD = 1.243 + 2.983 \times 10^{-3} (sa) + 4.187 \times 10^{-3} (sa) - 6.208 \times 10^{-2} (oc)$ | (Huf Dos Reis et al., 2024) |

$MBD_1$ = mineral bulk density (1.64 g cm$^{-3}$), VMF = bulk density of the mineral fraction per texture class in g cm$^{-3}$ according to the belgian texture triangle, depth = mean depth of the soil sample (cm), $K_1$ = 0.223 g cm$^{-3}$, $K_2$ = 1.27 g cm$^{-3}$, sa = sand (%), oc = organic carbon (%), om = organic matter (%), cl = clay (%), si = silt (%)



## 2.4 Machine Learning Algorithms

In this work, three types of machine learning algorithms were tested for BD estimation, neural networks (ANN, DNN), ensemble models (RF, XGBoost, LightGBM), and Support Vector Regression (SVR). Each was evaluated to identify the most
accurate and robust predictive model.

### 2.4.1 Artificial neural network (ANN)

ANN is a machine learning algorithm that mimics the structure of biological neurons, enabling computers to learn similarly to human brains. The commonly used ANN type is the multi-layer perceptron, which operates on a feed-forward model, processing inputs through neurons to produce outputs (Ghaderi et al., 2019). It consists of three layers: an input layer, hidden
layers, and an output layer (Liu et al., 2020). The performance of an ANN heavily depends on the careful tuning of its hyperparameters, which govern the model's structure and learning process. Key hyperparameters include the number of neurons in the hidden layers, which defines the model's capacity to learn complex patterns, and the activation function, such as ReLU, which introduces non-linearity to enhance learning capabilities. The learning rate plays a critical role in controlling the speed of weight updates during training, balancing stability and convergence efficiency. Batch size determines how often
the model updates weights during training, influencing both computational efficiency and generalization. Finally, the number of epochs dictates the number of times the model processes the entire dataset, requiring a balance between adequate training and computational resources. These hyperparameters, when appropriately tuned, significantly enhance the model's ability to learn and generalize, making them essential for achieving optimal performance.

### 2.4.2 Deep neural network (DNN)

DNN model was chosen for its ability to capture complex relationships in environmental and spectral features (Kim et al., 2023). DNNs typically consist of an input layer, multiple hidden layers, and an output layer. The increased depth of hidden layers enables DNNs to learn hierarchical feature representations, making them highly effective for tasks requiring advanced modeling capabilities. The performance of a DNN is significantly influenced by its hyperparameters, which control the architecture and training process. Key hyperparameters include the number of dense units in each hidden layer, which defines
the network's capacity to learn from data, and the activation function, such as ReLU, which enhances the model's ability to capture non-linear relationships. Dropout rates are critical for regularization, reducing the risk of overfitting by randomly deactivating neurons during training. Batch normalization is another important technique that normalizes layer inputs, stabilizing and accelerating the training process. Additionally, the learning rate determines the step size during weight updates, balancing convergence speed and stability. Callbacks, such as early stopping and learning rate reduction, play a pivotal role in
dynamically adjusting the training process to avoid overfitting and optimize performance. These hyperparameters collectively ensure that the DNN effectively learns and generalizes, making them essential to achieving accurate and reliable predictions for soil bulk density estimation.





### 2.4.3 Random Forest (RF)

RF algorithm, developed by Breiman (2001) and discussed by Hengl et al. (2018), builds numerous decision trees during
training. RF effectively manages outliers, handles unbalanced data distributions, accommodates non-linear patterns, and
captures complex relationships (Ao et al., 2019). The performance of RF is governed by several important hyperparameters.
The number of trees ($n\_estimators$) controls the size of the forest, with more trees generally improving stability and accuracy
at the cost of computation time. The maximum depth of each tree ($max\_depth$) determines the level of detail captured by the
model, while the minimum samples required to split a node ($min\_samples\_split$) and the minimum samples required to be at
a leaf node ($min\_samples\_leaf$) control the granularity of tree splitting. Another critical parameter is the $mtry$ value, which
specifies the number of features to consider at each split, balancing model performance and computational efficiency (Kesbi
et al., 2016). The final predictions are the weighted average of the individual tree's outputs.

### 2.4.4 Extreme gradient boosting (XGBoost)

XGBoost utilizes an ensemble learning approach and a gradient boosting framework, demonstrating robust performance in
handling complex datasets (Liu et al., 2024). This makes it a suitable choice for predicting BD with high-dimensional features.
XGBoost works by sequentially adding decision trees to minimize the residual errors of the previous models, effectively
improving prediction accuracy with each iteration. Key hyperparameters play a crucial role in optimizing XGBoost's
performance. The learning rate ($\eta$) determines the contribution of each tree to the final prediction, with smaller values ensuring
stable convergence. The number of trees ($n$ estimators) balances the trade-off between model complexity and computation
time. The maximum depth of each tree ($max$depth) controls the level of granularity in capturing data patterns, while the
minimum child weight parameter prevents overfitting by limiting the number of observations a leaf node can have.
Subsampling parameters, such as subsample and colsample_bytree, reduce overfitting by introducing randomness in the
training process. Collectively, these hyperparameters allow XGBoost to efficiently capture intricate patterns in high-
dimensional data while maintaining model robustness.

### 2.4.5 Light gradient boosting machine (LightGBM)

LightGBM model was utilized for predicting BD based on environmental and multispectral indices. The LightGBM model
demonstrated promising performance in predicting soil BD, leveraging its gradient-based boosting technique to handle
complex datasets efficiently and accurately. As well as, it can significantly outperform XGBoost in terms of computational
speed and memory consumption (Ke et al., 2017). The model's performance is driven by key hyperparameters. The learning
rate ($\eta$) governs the pace of weight updates during training, ensuring stable convergence. The number of trees ($n$ estimators)
controls the model's overall complexity and accuracy. The maximum number of leaves ($num\_leaves$) determines the
granularity of tree splits, directly influencing the model's ability to capture complex patterns. The minimum child samples





parameter ensures that leaves do not overfit small, noisy subsets of data. These hyperparameters allow LightGBM to deliver accurate predictions while maintaining computational efficiency, making it highly suitable for large-scale predictive tasks.

### 2.4.6 Support vector regression (SVR)

SVR was developed in was developed in the mid-1990s as an extension of the Support Vector Machine (SVM) algorithm (Müller et al., 1997). The goal of SVR is to approximate the relationship between input and output variables while minimizing prediction error (Yan et al., 2018). SVR performance relies on key hyperparameters: the kernel function captures non-linear patterns, the regularization parameter ($C$) balances training error and model simplicity, the kernel coefficient ($\gamma$) determines data point influence, and the epsilon parameter sets the tolerance margin for prediction errors. To ensure consistency, features were standardized using a StandardScaler to normalize input data. This combination of hyperparameter tuning and preprocessing enables SVR to deliver accurate and robust predictions while effectively managing noise and data variability.

### 2.5 Hyperparameter Tuning and Model Optimization

In this study, the Expected Improvement (EI) acquisition function was employed as part of the Bayesian Optimization (BO) process to fine-tune hyperparameters for all machine learning models, including ANN, DNN, RF, XGBoost, LightGBM, and SVR. The EI function, defined as Eq. (4),

$$EI(X) = \mathbb{E}[max(f(X_{best}) - f(X_{next}), 0)], \tag{4}$$

where $EI(X)$ is the Expected Improvement, $\mathbb{E}$ is the expected value, $X_{best}$ is the hyperparameter configuration with the best-observed objective function value so far, $f(X_{next})$ is the predicted value of the objective function at a new hyperparameter configuration $X_{next}$. By maximizing the EI, the BO process effectively identifies promising regions in the hyperparameter space where the Gaussian Process (GP) predictions exhibit high uncertainty (exploration) and simultaneously refines regions already known to perform well (exploitation) (Zhao et al., 2024). This iterative approach minimizes the risk of being trapped in local minima, ensuring a systematic and efficient improvement of the objective function. BO combined with k-fold cross-validation (k-FCV) was utilized as the hyperparameter tuning strategy to ensure optimal performance across all machine learning models (Mockus, 2005). BO efficiently explores the hyperparameter space by constructing a probabilistic surrogate model, to approximate the objective function (Sreenivasulu and Rayalu, 2024). The objective is to iteratively identify the hyperparameter configuration that minimizes the model validation loss, as defined in Eq. (5),

$$f(\theta) = \frac{1}{k}\sum_{i=1}^{k} \mathcal{L}_i(\theta), \tag{5}$$

where $f(\theta)$ is objective function, $k$ is the number of folds in the k-fold cross-validation, $\mathcal{L}_i$ is the loss function for the $i$-th fold of cross-validation, $\theta$ is the hyperparameters.

To investigate the impact of training methods, batch size versus entire dataset usage, two neural network models were designed and compared. The ANN model utilized mini-batches during training, where BO fine-tuned key hyperparameters, including



hidden layer configurations ([128, 64], [64, 32]), learning rates (0.001–0.01), and batch sizes (32, 64, 128). Each configuration
was validated using 10-fold cross-validation to ensure robustness. In contrast, DNN model was trained using the entire dataset
without batching. For DNN, BO optimized dense layer configurations (64 units per layer), dropout rates (0.1–0.5), and learning
rates (0.001–0.01). Similar to ANN, k-FCV was employed to validate the hyperparameter configurations. This experimental
setup allowed for a systematic evaluation of the effects of training methods on the predictive performance of the models.

For ensemble models, BO fine-tuned RF parameters, including the number of trees ($n\_estimators$ from 50-500) and splitting
criteria ($max\_depth$ from 5 to None, $min\_samples\_split$ from 2-10), validated through k-FCV. XGBoost, BO optimized
*learning rates* (0.01–0.1), *maximum depths* (5–20), and *subsampling ratios* (0.6–0.9). Similarly, LightGBM hyperparameters
were tuned for *learning rates* (0.01–0.1), *number of leaves* (20–80), and *min_child_sample* (10–30).

For SVR, BO refined kernel-specific parameters, including the regularization parameter ($C$ from 1-1000), kernel coefficient
($gamma$: 0.001-0.1), and *epsilon* (0.01–0.1). The integration of BO with k-FCV rigorously validated each configuration,
enhancing the generalizability and predictive performance of all models.

Each hyperparameter configuration was evaluated using 10-fold cross-validation, where the dataset was partitioned into 10
equal folds. For each fold, the model was trained on $k-1$ folds and validated on the remaining fold, iteratively cycling through
all folds to ensure a comprehensive evaluation. This process allowed the network to be trained and tested ten times, ensuring
robust performance assessment and reliable optimization of the hyperparameters (Wong and Yeh, 2019; Rodriguez et al.,
250 2009).

## 2.6 Validation and Model Assessment

The validation and model assessment process are divided into two parts: machine learning model development and the
validation of 76 PTFs. For the machine learning model development, the dataset is split into 80% for training and 20% for
testing. This split ensures that the model is trained on a substantial portion of the data while reserving an independent test set
for unbiased evaluation of its predictive performance. The training phase employs 10-fold cross-validation to tune model
parameters and assess the model's generalizability to unseen data. Evaluation metrics including the coefficient of determination
(R²), root mean squared error (RMSE) and mean absolute error (MAE) are applied as defined in Eq. (6)-(9). For the validation
of the 76 PTFs, the same metrics (R², RMSE, and MAE), along with the unbiased root mean square deviation (ubRMSD), are
utilized as defined in Eq. (6)-(10),

$$R^2 = 1 - \frac{\sum_{i=1}^n (X_i - Y_i)^2}{\sum_{i=1}^n (\bar{Y} - Y_i)^2}, \tag{6}$$

$$MSE = \frac{1}{n}\sum_{i=1}^n (X_i - Y_i)^2, \tag{7}$$

$$RMSE = \sqrt{MSE}, \tag{8}$$

$$MAE = \frac{1}{2}\sum_{i=1}^n |X_i - Y_i|, \tag{9}$$



$$\text{ubRMSD} = \sqrt{\frac{1}{2}\sum_{i=1}^{n}[(X_i - \bar{X}) - (Y_i - \bar{Y})]^2} \, , \tag{10}$$

where $X_i$ is the predicted values from the model, $Y_i$ is the observed values, $\bar{X}$ is the mean of the predicted values, $\bar{Y}$ is the mean of the observed values, $n$ is the total number of observations.

## 2.7 Model Implementations and Comparative Analysis

The model implementation focused on selecting the approach that provided the highest accuracy, lowest RMSE, and balanced feature importance using remote sensing data. The selected model was then applied to new datasets collected in 2009, utilizing

only OC data as the sole predictor for BD, as no ground-truth BD measurements were available for validation in that year. This approach allowed us to evaluate the model's robustness in estimating BD using limited inputs while ensuring its generalizability to new datasets. The 2009 dataset comprised 76,089 soil samples, containing OC percentages at a depth of 30 cm, which were integrated with remote sensing data to generate BD predictions and analyse temporal changes across diverse soil and land use conditions.

## 2.8 Uncertainty and Variability Quantification

The temporal uncertainty and variability of BD predictions between 2004 and 2009 were quantified using statistical techniques, including descriptive statistics, percentage change analysis, and hypothesis testing. Descriptive statistics were employed to calculate the mean and standard deviation for both years, providing insights into central tendencies and dispersion. Temporal changes were assessed by computing the percentage change in mean values and the percentage change in standard deviation,

while uncertainty was quantified as the absolute difference in standard deviations between the two periods (Faber, 1999). To evaluate the statistical significance of mean differences, Welch's t-test was applied as a robust method to compare the two years (Delacre et al., 2017). Below is a detailed description of the steps and formulas used following Eq. (9)-(12),

$$\Delta\mu \,(\%) = \frac{\mu_{2009} - \mu_{2004}}{\mu_{2004}} \times 100, \tag{9}$$

$$\Delta\sigma(\%) = \frac{\sigma_{2009} - \sigma_{2004}}{\sigma_{2004}} \times 100, \tag{10}$$

$$U = |\sigma_{2009} - \sigma_{2004}|, \tag{11}$$

$$t = \frac{\bar{X}_1 - \bar{X}_2}{\sqrt{\frac{s_1^2}{N_1} + \frac{s_2^2}{N_2}}}, \tag{12}$$

where $\mu_{2009}$ and $\mu_{2004}$ are the mean BD in 2009 and 2004, $\mu_{2009}$ and $\sigma_{2004}$ are the standard deviation BD in 2009 and 2004, $\bar{X}_1$ and $\bar{X}_2$ are the means of the two samples, $s_1^2$ and $s_2^2$ are the variances of the two samples, $N_1$ and $N_2$ are the sample sizes of the two samples.



## 3 Results

### 3.1 Descriptive Statistics

Table 3 summarizes the soil dataset. BD values range from 0.07 to 1.55 g cm⁻³, with an average of 1.28 g cm⁻³ and a standard deviation of 0.01 g cm⁻³. The data is negatively skewed (-2.91) with high kurtosis (16.30), indicating most values are clustered around the mean with a few lower extremes. OC content varies from 0.12% to 26.66%, averaging 1.83% and a standard deviation of 2.51%. High skewness (7.26) and kurtosis (65.88) suggest a distribution dominated by lower values with a few high outliers. The table details the 2004 soil properties: sand (0.30%-98%), silt (0.50%-83.50%), and clay (0.50%-84%). Sand has a near-normal distribution, while silt and clay show moderate skewness and kurtosis. High variability in OC and OM highlights the need for advanced modelling to capture these patterns accurately.

Figure 1(b) shows the USDA Soil Texture Triangle for the 2004 soil samples, revealing a predominance of clay and clay loam textures. Most samples are concentrated in high-clay content regions, indicating a significant presence of clay-rich soils, while smaller clusters in the sandy clay loam and loam categories reflect the textural variability across the study area.

**Table 3** Description of statistical analysis of soil data sampling in 2004 for 76 PTFs validation and developing BD predictive model

| Soil properties | Unit | Min | Mean | Max | SD | Median | Skewness | Kurtosis |
|:---:|:---:|:---:|:---:|:---:|:---:|:---:|:---:|:---:|
| sand | % | 0.30 | 42.91 | 98 | 28.79 | 44.85 | 0.04 | -1.35 |
| silt | % | 0.50 | 34.42 | 83.50 | 17.22 | 33.00 | 0.36 | -0.34 |
| clay | % | 0.50 | 22.68 | 84.00 | 20.35 | 15.00 | 0.99 | -0.17 |
| OC | % | 0.12 | 1.83 | 26.66 | 2.51 | 1.31 | 7.32 | 65.72 |
| OM | % | 0.21 | 3.13 | 45.96 | 4.31 | 2.25 | 7.32 | 65.72 |
| BD | g cm⁻³ | 0.07 | 1.28 | 1.55 | 0.01 | 1.31 | -2.96 | 16.55 |

The correlation analysis between the features and the target variable reveals that OC has the strongest correlation with BD, showing a very strong negative relationship (r = -0.92), indicating higher OC is associated with significantly lower BD, as shown in Figure 2. Other features exhibit much weaker correlations, both positive and negative. Positive correlations include CI (0.19), NDVI (0.16), and EVI (0.16), suggesting higher values in these indices are associated with higher BD. Negative correlations are observed with rainfall (-0.02), temperature (-0.02), aspect (-0.04), and elevation (-0.09), though their impact is minimal compared to OC. This highlights that while multiple features influence BD, OC is the most significant factor.



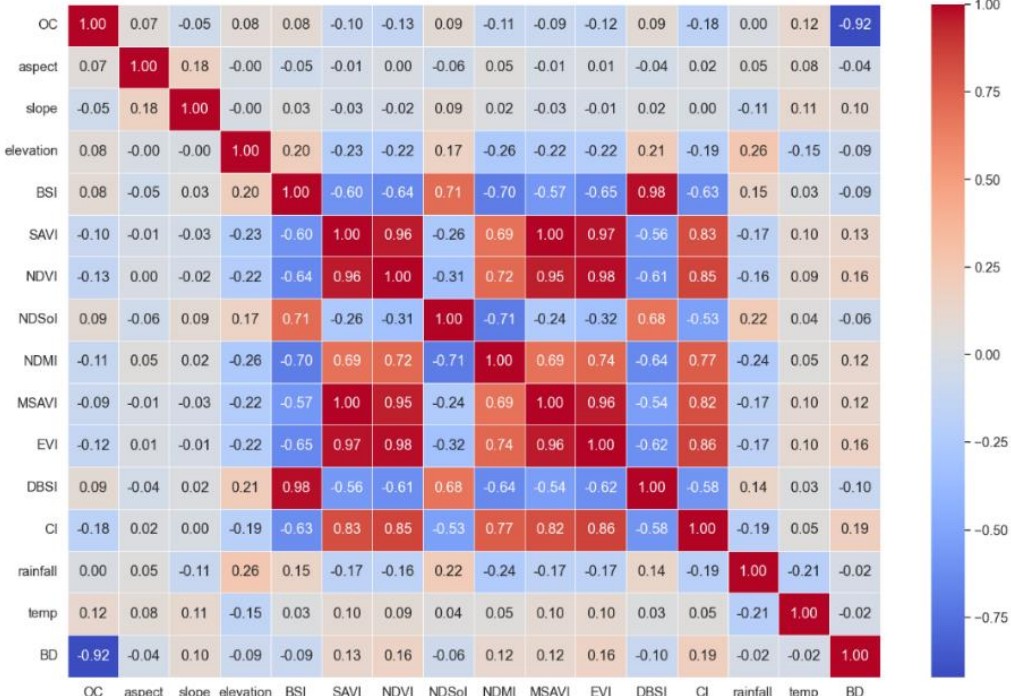

**Figure 2** Correlation matrix of BD predictors (remote sensing and environment variables) and BD.

### 3.2 Performance evaluation of 76 PTFs using in-situ data

The comparative evaluation of 76 PTFs was carried out using RMSE, using 236 in-situ soil datasets collected in 2004. The RMSE are illustrated in Fig. 3, showcasing the significant variability in predictive accuracy among the PTFs evaluated. The RMSE values ranged from 0.051 g cm⁻³ to 6.273 g cm⁻³, with a median of 0.171 g cm⁻³. The mean RMSE was 0.425 g cm⁻³, indicating that while many models performed adequately, there was considerable deviation among them. About 25% of the models achieved RMSE values below 0.101 g cm⁻³, showcasing strong predictive performance. The OC16 model exhibited

the highest performance, achieving an RMSE of 0.021 g cm⁻³, a MAE of 0.014 g cm⁻³, and an R² of 0.985, reflecting its superior accuracy and consistency in soil bulk density prediction. In contrast, the poorest-performing model, PSOC8, exhibited an RMSE of 6.273 g cm⁻³, highlighting significant predictive errors (Fig. 4). The MAE values, which measure the average magnitude of errors, ranged from 0.014 g cm⁻³ to 2.105 g cm⁻³, with a median value of 0.136 g cm⁻³. This suggests that half of the models had error magnitudes below this threshold. The ubRMSD values, representing unbiased random errors, varied

from 0.007 g cm⁻³ to 6.206 g cm⁻³, with a median of 0.0945 g cm⁻³, indicating that some models exhibited very low random errors while others had significant discrepancies.





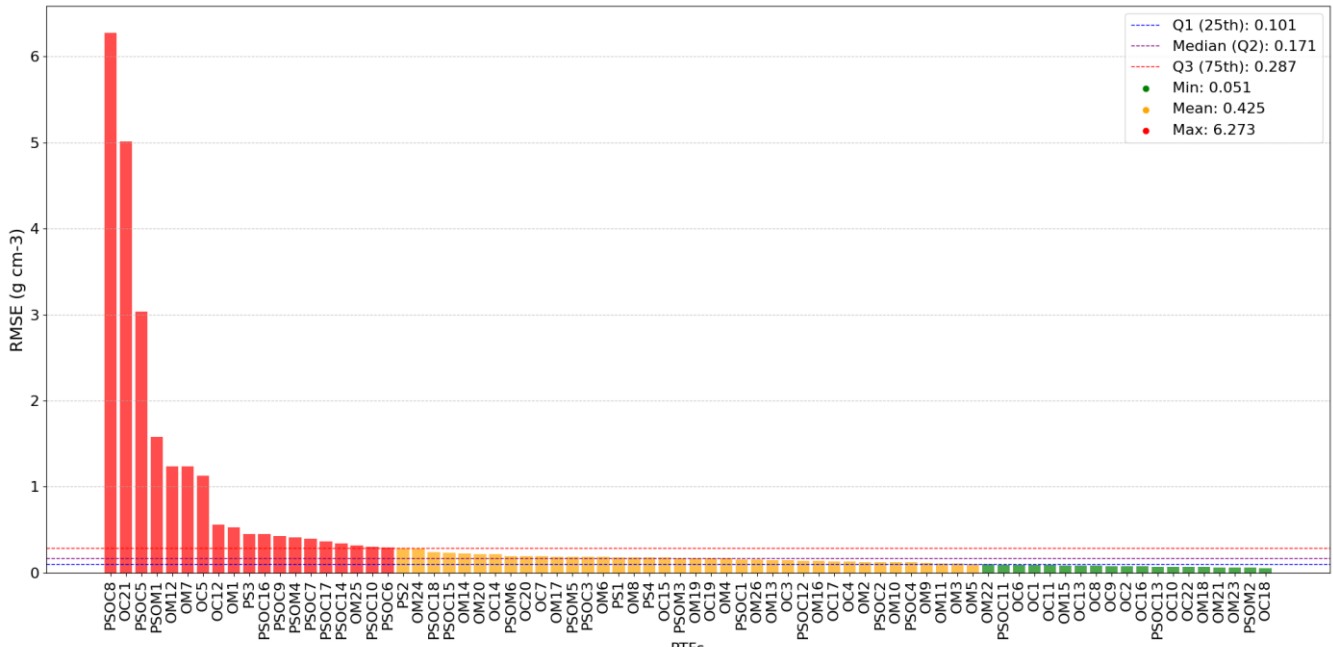

**Figure 3** RMSE comparison of 76 PTFs methods for soil bulk density estimation in 2004

The performance of predictive PTFs models for BD, as categorized by the parameters, exhibited significant variability across different groups (Table 4). The OC18 model demonstrated exceptional predictive capability, achieving the lowest RMSE of 0.051 g/cm³ and the highest R² of 0.885, making it the most accurate model overall. Conversely, the PSOC8 model showed the poorest performance, with the highest RMSE of 6.273 g/cm³ and an R² of -1328.22, reflecting substantial prediction errors and uncertainty. Among OM-based models, OM21 and OM23 stood out with RMSE values of 0.062 g/cm³, highlighting their

superior accuracy within the group. For PS-based models, PS4 the most reliable, with an RMSE of 0.179 g/cm³, though PS models generally exhibited higher variability and lower predictive reliability. In the PS_OM group, PSOM1 achieved the best performance, with an RMSE of 0.061 g/cm³, while in the PSOC group, PSOC13 performed strongly, with an RMSE of 0.072 g/cm³. Overall, models incorporating OC predictors, particularly OC18, consistently exhibited the highest accuracy and lowest uncertainty, whereas particle-size-based models were more prone to variability and less reliable in predicting BD.

**Table 4** Top three performing PTFs for BD estimation categorized by predictor groups

| PTFs | PTF Method | RMSE | ubRMSD | MAE | $R^2$ |
|---|---|---|---|---|---|
| Organic carbon based PTFs | OC18 | 0.051 | 0.041 | 0.014 | 0.885 |
| | OC22 | 0.072 | 0.053 | 0.055 | 0.824 |
| | OC10 | 0.072 | 0.071 | 0.056 | 0.824 |
| Organic matter based PTFs | OM21 | 0.063 | 0.052 | 0.046 | 0.867 |



| | | | | | |
|---|---|---|---|---|---|
| | OM23 | 0.063 | 0.052 | 0.046 | 0.867 |
| | OM18 | 0.070 | 0.053 | 0.050 | 0.835 |
| Particle size based PTFs | PS4 | 0.179 | 0.165 | 0.134 | -0.078 |
| | PS1 | 0.180 | 0.177 | 0.125 | -0.094 |
| | PS2 | 0.285 | 0.170 | 0.230 | -1.753 |
| Particle size and organic carbon based PTFs | PSOC13 | 0.072 | 0.067 | 0.041 | 0.824 |
| | PSOC11 | 0.094 | 0.092 | 0.067 | 0.701 |
| | PSOC4 | 0.120 | 0.118 | 0.088 | 0.514 |
| Particle size and organic matter based PTFs | PSOM2 | 0.061 | 0.057 | 0.041 | 0.874 |
| | PSOM1 | 0.162 | 1.567 | 0.111 | 0.110 |
| | PSOM3 | 0.171 | 0.067 | 0.159 | 0.008 |

### 3.3 Performance of ML-BD Predictive Model

To determine the most effective machine learning model for BD prediction, six algorithms were evaluated: ANN, DDN, RF, XGBoost, SVR, and LightGBM. The performance of each model was measured using the RMSE, MAE, and $R^2$ on both training and testing datasets, with hyperparameters tuned to achieve the highest accuracy (Table 6). Among the models, the ANN model achieved the highest predictive accuracy, with a testing $R^2$ of 0.986, the lowest RMSE of 0.017 g cm$^{-3}$, and an MAE of 0.012 g cm$^{-3}$, indicating its robustness in capturing the complex patterns in the dataset. The RF model also demonstrated strong performance, with a testing $R^2$ of 0.965, RMSE of 0.029 g cm$^{-3}$, and a minimal MAE of 0.010 g cm$^{-3}$, making it another reliable option for BD prediction. The XGBoost model, despite its superior performance on the training set ($R^2$ = 0.998, RMSE = 0.005 g cm$^{-3}$), showed a drop in accuracy on the testing set, with a testing $R^2$ of 0.936 and a higher RMSE of 0.040 g cm$^{-3}$. This suggests potential overfitting, where the model performs exceptionally well on the training data but struggles to generalize to new data. Similarly, LightGBM exhibited satisfactory performance, with a testing $R^2$ of 0.894 and RMSE of 0.053 g cm$^{-3}$, making it a viable alternative. The DNN model had a testing $R^2$ of 0.859, RMSE of 0.049 g cm$^{-3}$, and MAE of 0.035 g cm$^{-3}$, indicating moderate predictive power. In contrast, the SVR model recorded the lowest predictive accuracy, with a testing $R^2$ of only 0.549, RMSE of 0.116 g cm$^{-3}$, and the highest MAE of 0.074 g cm$^{-3}$, making it less effective for BD estimation in this context.

**Table 5.** The optimal hyperparameters

| Model | Architecture/Parameters | Optimizer | Loss Function | Training Parameters |
|---|---|---|---|---|
| ANN | Input: 128 neurons; Hidden: [64, 32] neurons; ReLU activation; Output: 1 neuron | Adam | MSE | 1000 epochs, batch size = 64 |
| DNN | Dense layers: [64, 64] neurons; ReLU activation; Batch normalization; Dropout regularization | Adam (lr-0.01) | MSE | 1000 epochs, early stopping, learning rate reduction |
| RF | *n estimators* =100; *max depth* = None; *min samples_split* = 4; *min samples_leaf*=1 | - | - | - |





| | | | | |
|---|---|---|---|---|
| XGBoost | *n estimators* =1000; *Max Depth*= 20; *Learning Rate* ($\eta$) = 0.01; *min child_weight*=2; *Subsample* = 0.8; *col sample* bytree=0.8 | - | - | - |
| LightGBM | *n estimators* =1000; *Learning Rate* = 0.01; *Number of Leaves* = 40 | - | - | - |
| SVR | Kernel: RBF; $C$=100; $\gamma$=0.1; $\epsilon$=0.1 | - | - | StandardScaler applied |

**Table 6.** Performance comparison of machine learning models for BD predictions

| ML models | Hyperparameters | Training | | | Testing | | |
|---|---|---|---|---|---|---|---|
| | | $R^2$ | RMSE | MAE | $R^2$ | RMSE | MAE |
| ANN | | 0.995 | 0.012 | 0.008 | 0.986 | 0.017 | 0.012 |
| DNN | | 0.900 | 0.030 | 0.050 | 0.859 | 0.049 | 0.035 |
| RF | | 0.986 | 0.020 | 0.004 | 0.965 | 0.029 | 0.010 |
| XGBoost | | 0.998 | 0.005 | 0.001 | 0.936 | 0.040 | 0.021 |
| SVR | | 0.922 | 0.048 | 0.041 | 0.549 | 0.116 | 0.074 |
| LightGBM | | 0.879 | 0.063 | 0.019 | 0.894 | 0.053 | 0.022 |

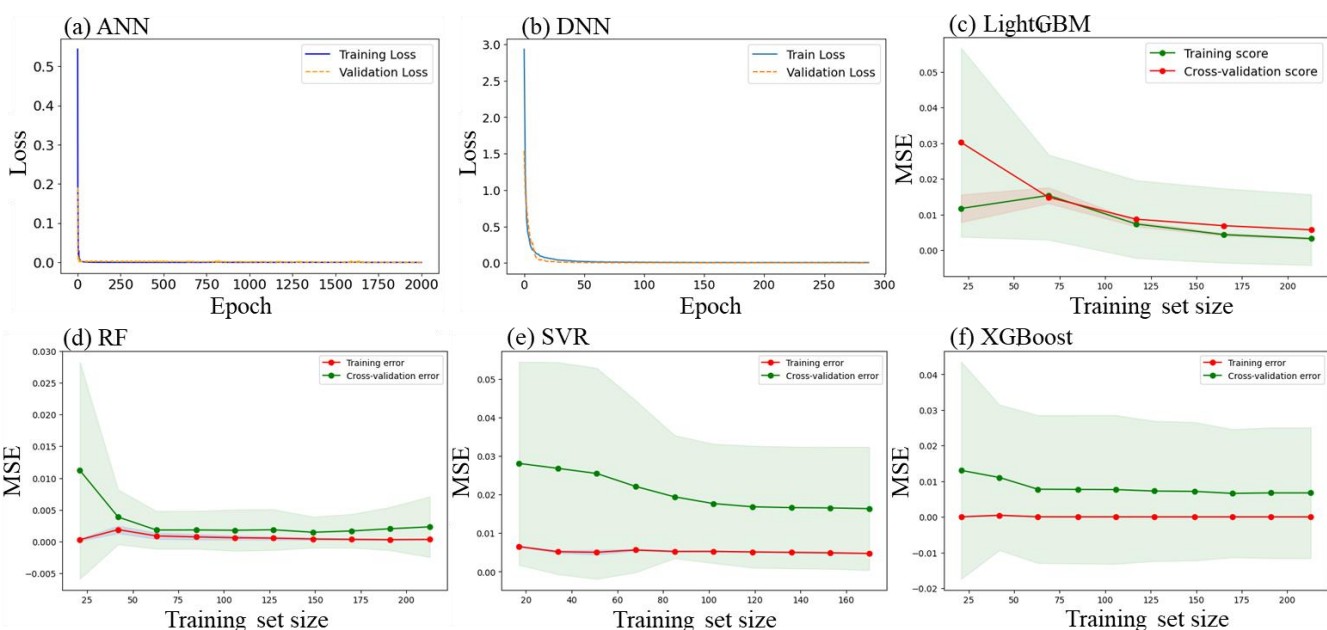





**Figure 4.** Loss function curves for neural network regression models: (a) Artificial Neural Network (ANN), (b) Deep Neural Network (DNN), and learning curves for other machine learning models: (c) LightGBM, (d) Random Forest, (e) Support Vector Regression (SVR), and (f) XGBoost for soil bulk density prediction.

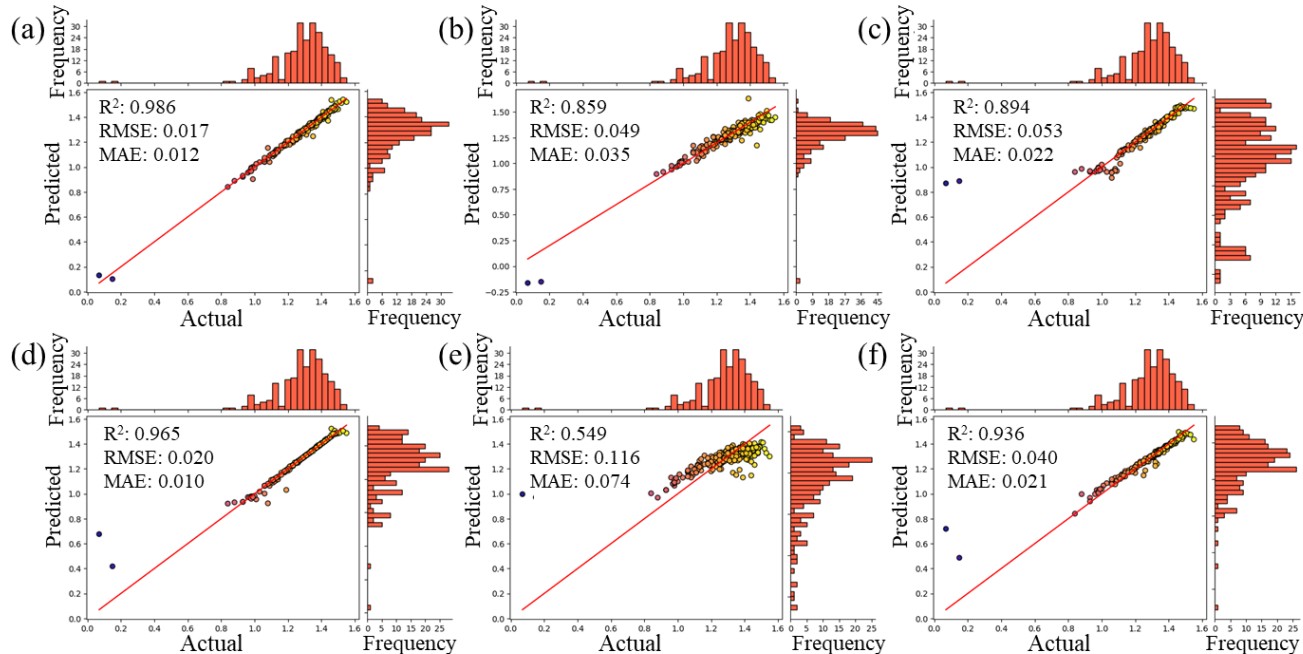

**Figure 5.** Scatterplot of predicted and actual BD using remote sensing data across difference machine learning: (a) ANN, (b) DNN, (c) LightGBM, (d) Random forest, (e) SVR, (f) XGboost

## 3.4 Importance of BD Predictors

The feature importance analysis was performed on six machine learning models (ANN, DNN, LightGBM, RF, SVR, and XGBoost) to evaluate the relevance of various remote sensing and environmental features in predicting BD. The analysis revealed distinct patterns in feature utilization across the models, highlighting the varying predictive strategies employed by each algorithm. The ANN model exhibited a relatively even distribution of feature importance, with temperature (7.18%) and slope (7.21%) being the most significant predictors. This suggests that the ANN model effectively integrates a broad spectrum of remote sensing and environmental data without heavily relying on a single feature. Similarly, the DNN model demonstrated a balanced feature utilization, identifying Bare Soil Index (BSI, 8.84%) and Dry Bare Soil Index (DBSI, 8.78%) as the primary predictors, indicating that it leverages a diverse set of inputs to achieve accurate BD predictions. In contrast, the LightGBM model showed a strong dependence on a limited set of key variables, with OC emerging as the dominant feature, accounting for 44.80% of its predictive power. Other relevant features included temperature (16.49%) and aspect (13.94%), reflecting a narrower focus compared to the ANN and DNN models. The RF model displayed an even higher dependency on OC, which



contributed to 90.54% of its predictive power, followed by temperature (8.52%). This heavy reliance on OC suggests that the RF model is highly sensitive to variations in organic carbon content, making it less effective in scenarios with limited or heterogeneous OC data. The SVR model, while also prioritizing OC at 54.49%, showed a more balanced distribution of secondary predictors, such as BSI (5.02%) and DBSI (4.80%), compared to the RF model. This indicates that while SVR is influenced by OC, it also considers other spectral indices, making it slightly more adaptable. The XGBoost model followed a similar pattern, with OC (50.25%) as the most important predictor, but also placed considerable weight on Clay Index (CI, 6.97%) and temperature (5.56%), reflecting a more diversified utilization of remote sensing data compared to LightGBM and RF.

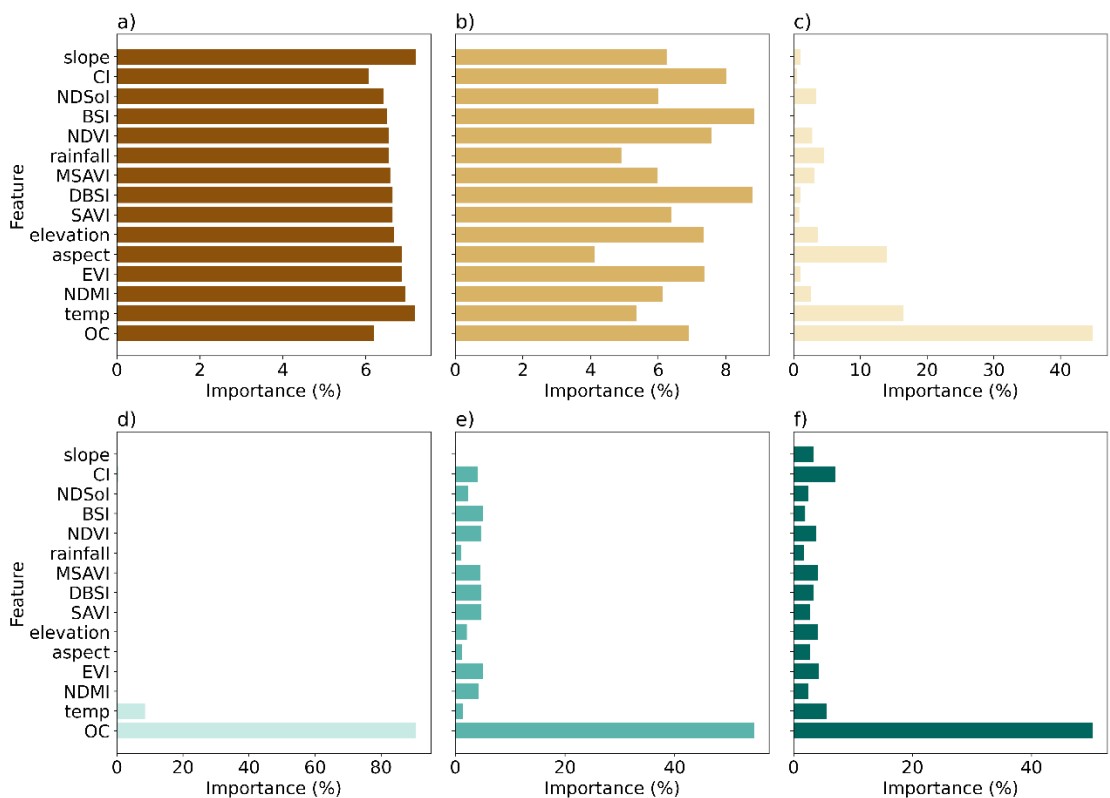

**Figure 6.** Variable importance for the six machine learning models in predicting BD across the study area: (a) ANN, (b) DNN, (c) LightGBM, (d) Random Forest, (e) SVR, (f) XGBoost.

## 3.5 Temporal Trends and Variability (2004–2009)

The temporal changes in BD between 2004 and 2009 using a robust ANN model, focusing on identifying significant shifts and evaluating associated uncertainties over time. Table 6 provides a summary of the descriptive statistics for the ANN model,





highlighting the changes in BD values, while Figure 9 visually represents these variations through boxplots and histograms. The results reveal a consistent increase in BD values from 2004 to 2009, suggesting a trend towards denser soils. The mean BD value rose from 1.28 g cm⁻³ in 2004 to 1.38 g cm⁻³ in 2009, reflecting a 7.27% increase. This increase may be attributed to factors such as intensified land management practices, reduced soil organic matter, or increased soil compaction over time.

The minimum BD values also showed a substantial increase, rising from 0.12 g cm⁻³ in 2004 to 0.95 g cm⁻³ in 2009. This shift indicates a decline in the prevalence of low bulk density soils, possibly due to reduced loose soil structures or enhanced soil consolidation. Additionally, the standard deviation of BD decreased significantly from 0.17 to 0.10 g cm⁻³, representing a 41.23% reduction in variability, which suggests that soil conditions became more uniform and stable over the study period. This trend may be associated with stabilized land use practices, reduced erosion, or more consistent soil management. Further

supporting these observations, the skewness improved from -2.81 in 2004 to -0.58 in 2009, while kurtosis decreased from 15.37 to -0.41, indicating a shift from a highly skewed and heavy-tailed distribution to a more symmetrical, normal-like distribution. This transformation suggests a reduction in the occurrence of extreme BD values and a more balanced distribution of BD by 2009.

The negative t-statistic of -8.4213 means that the mean BD in 2004 is lower than in 2009, showing a substantial difference between the two years. The large absolute value indicates strong evidence that this difference is not due to random chance.

The high degrees of freedom (235.5038) reflect that Welch's t-test correctly accounts for the unequal sample sizes and variances between the two datasets, ensuring reliable results.

Figure 8 provide additional insights into the frequency distribution of BD. In 2004, BD values displayed a broader spread with a peak around 1.30 g cm⁻³, while in 2009, the distribution narrowed and shifted towards 1.40 g cm⁻³, suggesting increased soil

compaction. The 2009 distribution also shows fewer low-density values, supporting the trend of more compacted soils over time. These temporal changes reflect a general increase in BD values and reduced variability, suggesting a transition towards denser and more consistent soil conditions across the study area. The lower uncertainty value in 2009 compared to 2004 indicates greater stability in the BD predictions, underscoring the robustness of the ANN model in capturing temporal dynamics in soil bulk density.

**Table 7.** Comparison of predicted BD values in 2004 and 2009 using ANN model compare with PTFs

| Year | Method | Min | Mean | Max | SD | Skewness | Kurtosis | Mean Change (%) | SD Change (%) | Uncertainty | Welch's t-test |
|------|--------|------|------|------|------|----------|----------|------|--------|-------------|----------------|
| 2004 | ANN | 0.12 | 1.28 | 1.55 | 0.17 | -2.83 | 15.73 | 7.274 | -41.23 | 0.07 | -8.42 |
| 2009 | ANN | 0.95 | 1.38 | 1.56 | 0.10 | -0.58 | -0.41 | | | | (235.50)* |
| 2004 | OC18 | -0.06 | 1.29 | 1.59 | 0.19 | -2.96 | 16.44 | 10.34 | -41.35 | 0.08 | -10.64 |
| 2009 | OC18 | 0.96 | 1.42 | 1.71 | 0.11 | -0.39 | -0.12 | | | | (235.50)* |

* is degree of freedom





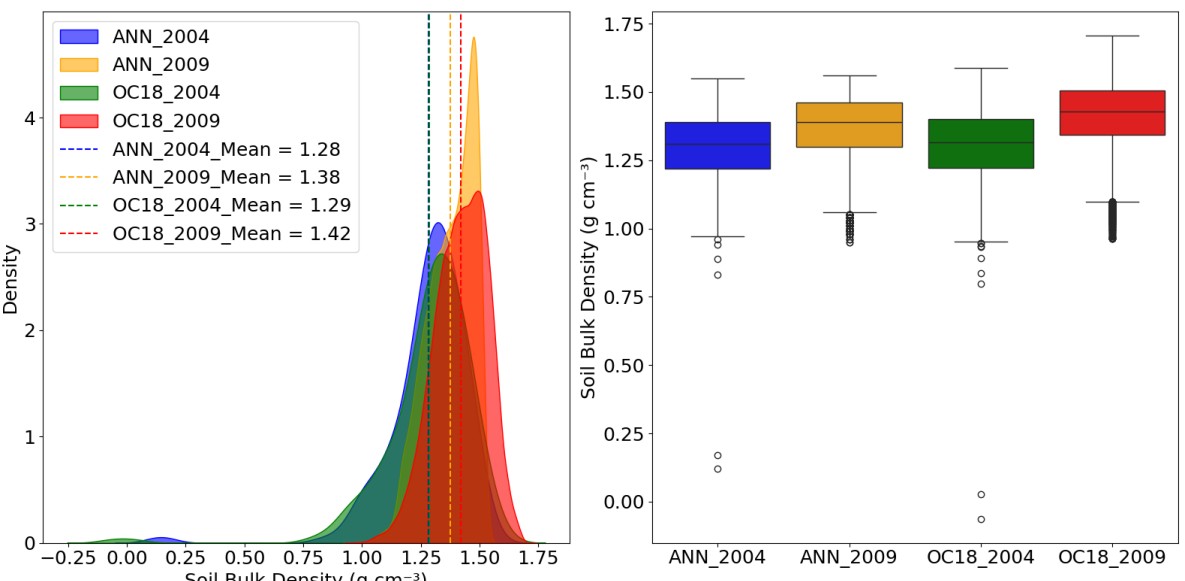

**Figure 7.** Histogram of predicted soil bulk density (BD) in 2004 and 2009 between a robust ANN model combined remote sensing and
PTFs (OC18) method.

## 3.6 Spatial distribution of soil bulk density (BD) in Thailand

Figure 9 illustrates the spatial distribution of BD across Thailand in 2009, categorized into six distinct BD classes ranging
from 0.95 g cm⁻³ to 1.56 g cm⁻³. The distribution patterns show a clear spatial heterogeneity, with a notable concentration of
lower BD values (0.95–1.27 g cm⁻³) predominantly located in the northern, central, and southern regions. These lower bulk
density values are often associated with less compacted soils, which could be linked to a combination of lower OC content and
reduced land management pressures.

Conversely, higher BD values (1.44–1.56 g cm⁻³) are primarily found in the northeastern part of Thailand, indicating more
compacted soils. This pattern suggests that the northeastern region might have undergone more intensive agricultural activities
or experienced land degradation processes, leading to increased bulk density. The presence of higher BD values in these areas
could pose challenges for crop productivity due to potential soil compaction and decreased soil porosity, which can impact
root penetration and water infiltration.



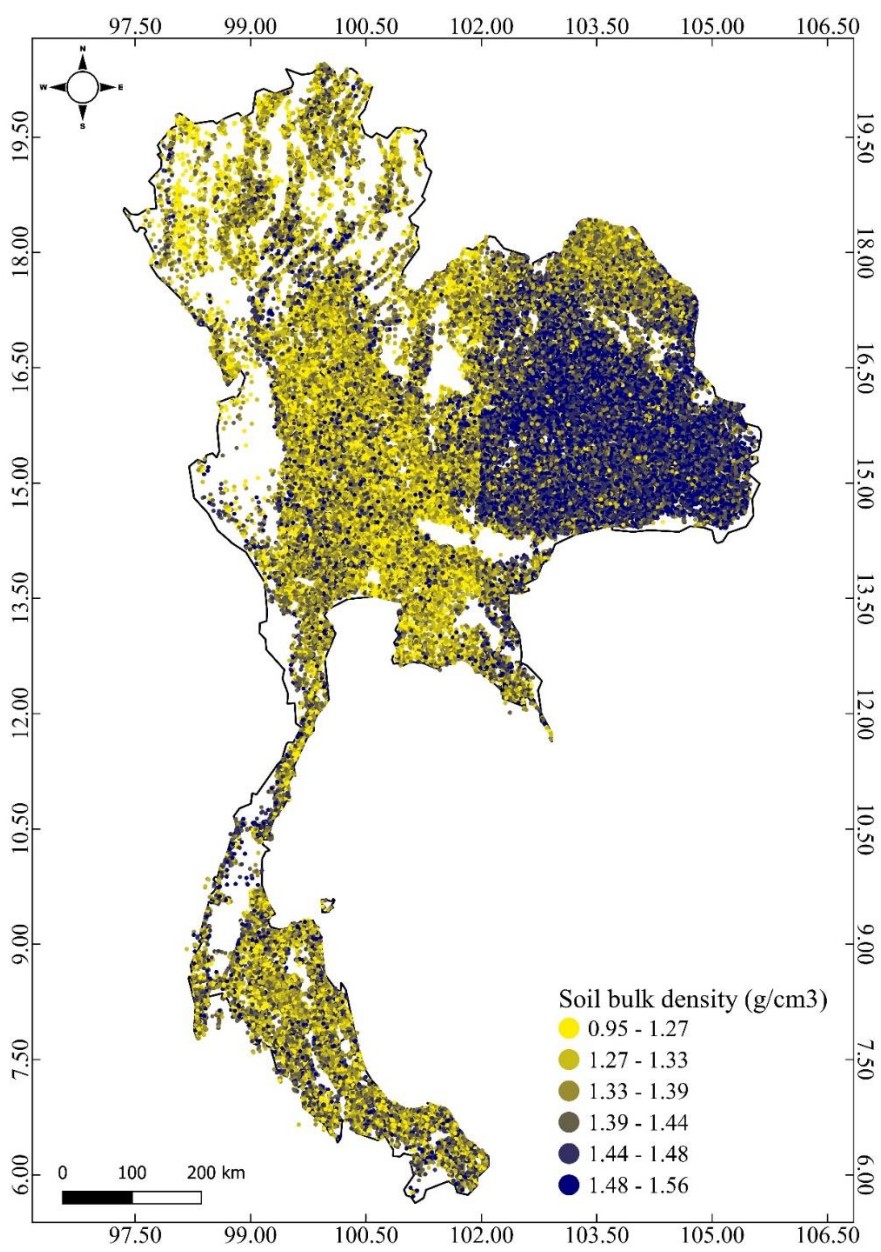

**Figure 8** Spatial distribution of BD in 2009





## 4 Discussion

### 4.1 Limitations of PTF-Based BD Models

The evaluation of BD predictions using 76 published PTFs revealed several limitations, primarily related to the narrow focus on specific soil properties. These models often rely heavily on OC or Particle Size (PS), which restricts their applicability across diverse landscapes. While OC-based PTFs have been shown to perform well under stable and homogeneous soil conditions (Alexander, 1980; Sevastas et al., 2018; Yi et al., 2016), their over-reliance on OC can result in significant deviations when soil properties are influenced by additional factors, such as soil texture, moisture, and topographic variations (Lupikis et al., 2017; Nasta et al., 2020; Tsui et al., 2013). Our findings show that OC-based PTFs, while exhibiting strong alignment with the RS-ANN model in mean BD values, displayed higher variability and greater prediction uncertainty, particularly in regions with fluctuating organic matter content. The heavy dependence on OC as the primary predictor of BD was evident in PTFs such as Manrique and Jones (1991), Reidy et al. (2016), and Do et al. (2024), which showed higher uncertainty. This aligns with studies by Rawls et al. (2004) and Honeysett and Ratkowsky (1989), who found that OC-based models tend to perform poorly in soils where other factors, such as soil compaction, soil texture, or land management practices, have a stronger influence on bulk density. Additionally, Saini (1966) and Williams (1971) demonstrated significant deviations even under low OC conditions (<2%), further emphasizing the limitations of OC-reliant PTFs in complex and heterogeneous landscapes (Vasiliniuc and Patriche, 2015). High sensitivity to extreme OC or OM values (>20%) was another key limitation observed in several PTF models, including Heuscher et al. (2005), Valzano F et al. (2005), and Tomasella and Hodnett (1998), which often resulted in negative BD predictions. This phenomenon is particularly problematic in soils with high organic matter content, where extreme BD values can lead to unreliable estimates (Sevastas et al., 2018). Our study identified 12 PTFs that were highly sensitive to OC values above 20%, frequently producing unrealistic or negative BD predictions. This issue has been highlighted in previous research, where Minasny and Hartemink (2011) and Jeffrey (1970) reported that PTFs derived from small datasets or regions with specific soil properties tend to perform poorly when applied outside their calibration range.

The analysis also revealed that PTFs based solely on PS or those incorporating PS and OC (PS-OC) showed substantial prediction errors and high variability compared to OC-based and remote sensing-integrated models. This finding is consistent with Al-Qinna and Jaber (2013), who found that PS-based PTFs perform well in sandy soils but struggle in regions with mixed soil textures. Models such as Akpa et al. (2016) and Bernoux et al. (1998), which were heavily reliant on particle size, displayed significant discrepancies in minimum and mean BD values when applied to heterogeneous landscapes. This variability suggests that PS-based models may not capture the complex interactions between soil properties, leading to reduced reliability in diverse regions. Our findings are supported by Nasta et al. (2020), who emphasized that PTFs based on static soil properties fail to capture temporal variability, leading to high uncertainty when applied over extended periods. Many of the evaluated PTFs displayed significant uncertainty when applied outside their calibration regions. Models such as Prevost (2004) and Hollis et al. (2012) showed relatively low uncertainty in regions with homogeneous soil properties but failed to perform reliably in more





complex environments with varying soil textures and climatic conditions. This is consistent with findings by Xu et al. (2016) and Vasiliniuc and Patriche (2015), who reported that PTFs developed using small or region-specific datasets tend to produce biased estimates when applied at larger scales. The limited generalizability of these models suggests that their use should be confined to areas with similar soil properties and environmental conditions to those in the calibration dataset.

## 4.2 Impact of BD Predictors and Organic Carbon for Model Prediction

The feature importance analysis for BD prediction revealed distinct patterns in predictor usage across various machine learning models, highlighting the critical role of different environmental and remote sensing variables in model performance. Each of the six models evaluated (ANN, DNN, LightGBM, RF, SVR, and XGBoost) exhibited unique dependencies on OC and other predictors, demonstrating the diverse strategies and strengths of these algorithms. The analysis identified three distinct patterns in feature utilization across models: (1) balanced use of diverse predictors in ANN and DNN models, (2) heavy reliance on OC in tree-based models.

The integration of remote sensing data significantly enhances the predictive performance of machine learning models for BD estimation. Memon et al. (2019) reported the ability of neural network algorithms to capture complex, non-linear relationships in high-dimensional data. The balanced utilization of temperature, NDMI, EVI, and slope by the ANN model indicates its strength in leveraging diverse environmental information for accurate BD predictions, supported by Schillaci et al. (2021), Yi et al. (2016), and Jalabert et al. (2010). This approach contrasts with other models like RF and SVR, which predominantly rely on a single predictor (OC), reducing their effectiveness in heterogeneous landscapes. Studies by (Schillaci et al., 2021) and (Yang et al., 2007) corroborate these findings, showing that tree-based models, such as RF and LightGBM, tend to overfit specific features, making them less robust in regions with complex soil properties. This reliance limits their capacity to generalize and highlights the advantage of ANN models in utilizing a comprehensive set of predictors for robust soil property estimation across diverse environmental conditions. The feature importance analysis of the ANN model revealed that slope, temperature, NDMI, and EVI were the top predictors, highlighting their significant influence on BD estimation. Each of these variables interacts with soil properties and land surface dynamics. Tsui et al. (2013) reported steeper slopes are often linked to lower SOC due to higher erosion rates and reduced water infiltration, leading to increased bulk density in topsoil layers. Additionally, slope affects soil compaction, root penetration, and the overall structure of the soil profile, influencing the spatial distribution of soil properties (Lupikis et al., 2017). Consistent with the Results of Yang et al. (2007) and Schillaci et al. (2021). Davidson and Janssens (2006), Jalabert et al. (2010) and Yi et al. (2016) demonstrated the effects of temperature on organic matter decomposition and soil compaction, while Gao (1996) validated the use of moisture indices in capturing soil moisture dynamics, which are closely linked to BD changes. Finally, Huete et al. (2002) and Galle et al. (2021) confirmed the utility of EVI and other vegetation indices in reflecting soil structure and health, supporting their inclusion in BD prediction models.





In contrast, models like Random Forest (RF) and Support Vector Regression (SVR), which relied predominantly on OC, showed less balanced feature importance and produced less generalizable outcomes. Previous studies have consistently reported a strong negative correlation between OC and BD (Yang et al., 2007; Tsui et al., 2013; Lupikis et al., 2017), emphasizing the role of OC as a key driver of soil bulk density. However, over-reliance on OC can lead to significant predictive errors in regions where other soil and environmental factors, such as texture, moisture, and topographic variations, play a larger role. Perie and Ouimet (2008) and Minasny and Hartemink (2011) highlighted that OC-based models may perform poorly in soils with varying mineral compositions or where land management practices strongly influence soil properties. This is consistent with the findings of Al-Qinna and Jaber (2013) who reported significant prediction errors when applied to diverse soil textures. The sensitivity of models to specific soil properties resulted in high variability and lower stability. The study by Xu et al. (2016) supports this observation, suggesting that incorporating particle size without a robust framework for integrating other soil characteristics can lead to increased uncertainty and limited applicability across diverse landscapes.

### 4.3 Assessing the adaptability of robust ANN model

The study evaluated the adaptability of an ANN model using BD data from 2004 to predict BD in 2009. Four key factors underscore the effectiveness of ANN models in BD prediction: 1) ANN autonomously learns and extract pertinent features from input data during training. This process allows ANN to capture complex patterns and relationships in data without explicit feature engineering from large datasets (Lecun et al., 2015), regularization techniques to bolster model generalization (Zhang et al., 2016). This approach allows ANN to effectively handle multidimensional data such as environmental and remote sensing indices (Figure 7(a)). Contrarily, Katuwal et al. (2020) encountered challenges and did not achieve satisfactory results when attempting to utilize only vis–NIR spectra for predicting BD.; 2) ANN adeptly capture nonlinear relationships inherent in BD data (Negiş, 2024; Erzin et al., 2008), crucial for accurate predictions across varying environmental conditions (Dragović, 2022) and agricultural lands (Abbaspour-Gilandeh et al., 2023).; 3) ANN ability to predict in the year 2004 and 2009 in the study area significantly enhances their reliability in BD prediction. Studies demonstrate ANN consistent performance over time (Abiodun et al., 2018), adapting well to datasets from different years or regions (Zhang, 2010). Unfortunately, the study by Hateffard et al. (2023) reported a poor result ($R^2$ = -0.746) when integrating ANN with NDVI and spectral bands.; 4) ANN demonstrate notable resilience to data variability and outliers, essential for maintaining stable predictions of BD (Khemis et al., 2022). This resilience enables effective handling of variations in soil characteristics and environmental conditions, enhancing reliability in long-term soil studies (Ünal et al., 2023). Despite previous studies by (Katuwal et al., 2020) and (Hateffard et al., 2023) failing to achieve accurate BD predictions, ANN have shown significant advancements in their predictive capabilities. In this study, we developed a robust ANN model for BD prediction using open-source remote sensing data, achieving high accuracy across different years. The effectiveness of ANN performance is supported by studies by (Li et al., 2013) and (Zhao et al., 2009).



## 4.4 Uncertainty and Variability in BD Prediction

This study evaluated the uncertainty and variability in BD predictions between 2004 and 2009 using the RS-based ANN model and traditional PTFs. Our findings revealed five key sources of uncertainty and variability that significantly impact the reliability of BD predictions, influencing their applicability across diverse landscapes. 1) Dependence on OC as the primary predictor emerged as a major source of uncertainty in both traditional PTFs and several ML models. Studies have shown that PTFs exhibit high sensitivity to OC leading to inconsistent predictions in soils with varying OC levels. This sensitivity can

result in overfitting (Mcbratney et al., 2003) and poor generalizability when applied across different soil types and landscapes (Hou et al., 2024). Similarly, ML models such as Random Forest (RF) and XGBoost, which also depend heavily on OC (Chen et al., 2024), demonstrated increased variability and instability under varying conditions. The integration of diverse predictors, such as spectral indices and topographic data, as employed in the RS-ANN model, reduced uncertainty, indicating higher stability and improved prediction reliability (Jain and Zongker, 1997). 2) Temporal variations were another key source of

uncertainty. While the RS-ANN model showed a moderate increase in mean BD, PTF models showed more drastic changes (11.01% and 10.34%, respectively). This discrepancy suggests that models relying heavily on OC tend to overestimate temporal changes in BD, resulting in less stable predictions over time. 3) Changes in standard deviation (SD) between 2004 and 2009 highlighted variations in spatial prediction accuracy. While the RS-ANN model showed a 41.23% reduction in SD, indicating more consistent BD values, other models experienced sharper decreases, raising concerns about potential overfitting.

Spatial variability in BD predictions can be influenced by differences in soil structure and management practices, which are not adequately captured by traditional PTFs. 4) Skewness and kurtosis analyses revealed that the RS-ANN model improved from a highly skewed distribution in 2004 (skewness = -2.81, kurtosis = 15.37) to a more balanced distribution in 2009 (skewness = -0.58, kurtosis = -0.41). In contrast, PTFs continued to show high skewness and kurtosis, indicating persistent prediction errors for outliers. This reflects a lack of robustness in handling extreme BD values, which are critical for accurate

soil assessments. 5) The RS-ANN model demonstrated broader applicability across diverse soil types and land uses compared to traditional PTFs and OC-dominant ML models. This adaptability is crucial in regions with heterogeneous soil properties, where multiple factors (e.g., soil moisture, texture, and topography) influence BD. Studies have shown that models integrating diverse input variables perform better in capturing complex soil dynamics and reducing prediction uncertainty.

## 5 Conclusion

This study developed a reliable BD prediction model by combining remote sensing data, environmental factors, and several machine learning techniques. The ANN model showed better performance than traditional PTFs and other machine learning models. The ANN model's balanced use of predictors effectively captured spatial and temporal variability in BD, offering more stable and reliable predictions with lower uncertainty compared to the OC-dependent PTFs and ensemble-based models. The temporal analysis from 2004 to 2009 revealed a consistent increase in mean BD values with reduced variability, indicating

the model's robustness for long-term monitoring. Additionally, the ANN model successfully represented the spatial

heterogeneity of BD across Thailand, providing critical insights for soil management and sustainable land-use planning. ANN model, leveraging remote sensing data, proves to be a valuable tool for national-scale BD estimation, enabling more precise soil health monitoring, carbon accounting, and sustainable land management. Future research should focus on refining the temporal dynamics of the model would allow for more accurate monitoring of long-term soil changes, especially under varying

climatic conditions and land-use practices. Integrating time-series analysis with climate projections could provide more accurate predictions of how BD and other soil properties evolve over time.

**Data availability**

The data and analyses that support these findings will be made available in response to a reasonable request but are not hosted in an online repository at this time in order to protect the privacy of growers.

**Author contributions**

SO: conceptualization, investigation, data curation, formal analysis, visualization, writing (original draft). ZS: writing (review and editing). AP: resources, writing (review and editing). ZA: resources, writing (review and editing).

**Competing interests**

The contact author has declared that none of the authors has any competing interests.

**Acknowledgements**

We thank to the Soil Research Survey and Research Division, Land Development Department, Thailand, for providing the valuable soil sampling data utilized in the development and validation of the soil bulk density estimation model.

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
