# Peer review of "Reducing Temporal Uncertainty in Soil Bulk Density Estimation Using Remote Sensing and Machine Learning Approaches"

_EGUsphere, 2025_

## Author Comment (AC1)

**Author Response to Reviewer Comments**

**1. Main Comments**

**Reviewer comment:**

"This paper presents an approach to estimate soil bulk density (BD) from soil data, environmental data, and remote sensing data. Classical pedotransfer functions (PTF) are compared with ML methods that include also remote sensing data to predict soil bulk density. An important result of the study is that soil organic matter or soil organic carbon are the most important input variables of PTFs and PTFs that use only these input variables perform the best. As a consequence, these PTFs relate changes in BD over time only to changes in soil organic matter or carbon. ML approaches also include other variables that could be linked to land use and land management. This improves the prediction of BD compared to the classical PTFs. But, to what extent these extra variables influence the BD estimates depends on the type of method that is used. A difference in sensitivity to different variables affects how predictions of changes in BD respond to changes of input variables over time. Comparing the predicted distributions of BD in 2009 with those of 2004, it seems that both PTFs and ML methods predict similar changes, although there are some differences. The importance or relevance of these differences was not very clear. Furthermore, since no measurements of BD in 2009 were available, it was not possible to verify whether changes in BD were predicted more accurately using the ML method. Given this lack of validation, "the authors should give other evidence that demonstrates the additional value of the ML approach they propose. For example, can the differences between the changes in BD that are predicted by the ML and PTF approaches be related to independent information on management etc… ? What is the correlation between the changes in BD that are predicted by the two approaches? Since the change in BD is probably small compared to its spatial variability, it would be interesting to know whether the two approaches predict similar spatial patterns of the change and how these patterns of change are related to which input variables." I think this additional information is needed to give the paper more relevance."

**Response:**

We sincerely thank the reviewer for the insightful and constructive feedback, which has helped to improve the clarity and scientific depth of our manuscript. In response, we have carefully revised the manuscript to address the four key concerns as follows:

**First,** regarding the influence of input variable sensitivity on BD predictions over time, we expanded explanations in Sections 3.4 and 4.2. These revisions clarify how the ML model incorporates remote sensing and environmental variables (e.g., NDVI, BSI, slope, temperature), allowing it to capture BD variation associated with land surface dynamics, unlike classical PTFs which rely solely on organic carbon. **Second,** we acknowledge the limitation that in-situ BD measurements for 2009 are unavailable. This is addressed in Section 5 (Conclusion). To provide additional evidence of the added value of the ML approach, we revised Section 3.6 and Figure 8 to more clearly illustrate the prediction

differences between the ANN model and PTFs. Notably, PTFs were unable to handle high organic carbon values, leading in some cases to negative or unrealistic BD estimates (Figure 8d). In contrast, the ANN model remained robust. We also included a correlation analysis of predicted BD between models (Figure 8c) to further visualize prediction differences. **Third,** as the 2004 and 2009 soil samples were collected at different geographic locations, it was not methodologically appropriate to compute point-wise BD changes (ΔBD) or correlations of change between models. Therefore, instead of direct spatial comparison, we evaluated the distributional characteristics of BD predictions in 2004 and 2009 (mean, SD, skewness) in Table 7 and Figure 7, and compared model robustness and sensitivity to input variability. These results confirmed that the ANN model yielded more stable and realistic outputs across years, while PTFs were more sensitive to extreme OC inputs. **Finally,** while spatial patterns of BD change between years could not be directly mapped due to the non-overlapping sample locations, we addressed the reviewer's concern by performing a temporal feature importance analysis (Section 3.6, Table 8). This comparison revealed shifts in predictor influence between 2004 and 2009, most notably, increases in the importance of rainfall and SAVI, and decreases in NDMI, slope, and MSAVI. This finding has also been incorporated into the abstract to highlight the ANN model's temporal sensitivity and broader interpretability relative to traditional PTFs.

[Figure]

**Figure 1** Comparison of soil bulk density (BD) predictions in 2009 using ANN with RS inputs (a) and PTF (OC18) (b), the Pearson correlation between ANN and PTF predictions (c), and the histogram of corresponding BD density distributions (d).

**Table 1** Temporal comparison of relative importance (%) of predictor variables in ANN-Based BD estimation models in 2004 and 2009

| No | Feature | Relative Importance in 2004 (%) | Relative Importance in 2009 (%) | Change (%) |
|----|---------|-------------------------------|-------------------------------|------------|
| 1 | BSI | 7.16 | 6.81 | -0.35 |
| 2 | NDSoI | 7.18 | 7.23 | +0.05 |
| 3 | NDVI | 6.75 | 7.22 | +0.47 |
| 4 | NDMI | 6.88 | 6.02 | -0.86 |
| 5 | SAVI | 6.23 | 6.89 | +0.66 |

| 6 | OC | 6.06 | 6.95 | +0.89 |
| 7 | DBSI | 7.05 | 6.86 | -0.19 |
| 8 | temp | 6.95 | 6.55 | -0.40 |
| 9 | elevation | 6.43 | 6.51 | +0.08 |
| 10 | rainfall | 5.95 | 7.51 | +1.56 |
| 11 | EVI | 6.88 | 6.55 | -0.33 |
| 12 | MSAVI | 6.98 | 6.44 | -0.54 |
| 13 | aspect | 6.18 | 6.42 | +0.24 |
| 14 | CI | 6.46 | 6.20 | -0.26 |
| 15 | slope | 6.86 | 5.83 | -1.03 |

**2. Detailed Comments**

**Ln 26: 'Surface BD is dominated factor' Change to Surface BD is a dominating factor…**
**Response:** Revised as suggested to "Surface BD is a dominating factor."

**Ln 31: A reference would be needed here.**
**Response:** We have added supporting references.

**Ln 32 `Pedotransfer Functions (PTFs) have long been used to estimate BD by predicting soil properties based on readily available soil attributes.` This sentence has a strange structure. Skip: by predicting soil properties.**
**Response:** We have revised.

**Ln 50: 'In contrast, vis–NIR spectra from spectroscopy did not show significant differences in performance compared to PTFs-based models, but were still superior (Katuwal et al., 2020).' This is contradictory.**
**Response:** We revised the sentence for clarity by deleting "but were still superior."

**Ln 58: 'leading to issues such as overestimation' I think this is one specific issue but not a general problem.**
**Response:** We agree with the reviewer and have revised the sentence to "…which in some cases may result in overestimation." for accuracy

**Ln 83: 'Additionally, soil samples with OC data collected in 2009 were used for model implementation.' How many.**
**Response:** We have clarified this in the manuscript by specifying that 76,089 soil samples with OC data were collected in 2009 and used for model implementation.

**Ln 84: 'These samples included measurements of' which samples? The ones collected in 2004 and in 2009?**

**Response:** We have clarified this in the manuscript to specify that the measurements refer to the 236 soil samples collected in 2004 for model development.

**Eqs 1 and 2 are nearly identical. Eq 1 can be skipped. Eq 3 is trivial and can be skipped as well.**

**Response:** We have removed Eq. 1 and Eq. 3 from the manuscript, keeping only Eq. 2 for clarity.

**Figure 1: the color scale of the histogram does not match with that in the figure.**

**Response:** We have revised Figure 1 to ensure a consistent color scale across all panels (map, texture triangle, and histogram) and added a clear color bar.

[Figure]

**Ln 83: temperate climate: shouldn't it be tropical climate?**

**Response:** We corrected to "tropical climate"

**Ln 111: weighted median. Which weights were used?**

**Response:** We clarified that no weighting was applied and used a pixel-wise median composite of all cloud-free images (January–December). The text has been revised by replacing "weighted median" with "median" for clarity.

**Ln 135: 'Root Mean Square Error (RMSE)' with respect to what? The 2004 BD measurements?**

**Response:** We confirm that RMSE was calculated with respect to the observed BD measurements from 2004 in order to validate the PTFs. The PTF with the lowest RMSE was then selected and applied to the 2009 dataset for comparison with the ML+RS model. To avoid confusion, we have revised the manuscript text to explicitly state that RMSE was calculated against the 2004 BD measurements.

**Ln 270 'as no ground-truth BD measurements were available for validation in that year' The main purpose of the study was to investigate if the change in BD over time could be derived using the PTFs. To my understanding, that would require sampling of BD over time.**

**Response:** We agree with the reviewer that the absence of ground-truth BD measurements in 2009 is a limitation of our study, as it prevents direct validation of temporal changes. To acknowledge this, we have revised the manuscript and added one sentence in the Conclusion section before the future work paragraph: "A key limitation of this study is the absence of ground-truth BD data in 2009, which restricted direct validation of temporal predictions." This addition makes the limitation explicit while emphasizing that future BD sampling will be necessary for validating long-term model predictions.

**Ln 272: The 2009 dataset comprised 76,089 soil samples, containing OC percentages at a depth of 30 cm. Were sites where samples were taken in 2004 revisited in the 2009 campaign?**

**Response:** We clarify that the 2004 and 2009 soil datasets were collected under different sampling campaigns and therefore do not correspond to exactly the same locations. However, both datasets cover the same soil series, soil texture groups, and land-use types, and were collected from sites located in close proximity wherever possible. This ensures a high level of comparability between the two campaigns despite the absence of exact site revisits.

**Ln 286: If you want to investigate changes of a variable in time, you best observe the parameter at the same location. Then you do a paired t-test.**

**Response:** We agree with the reviewer that paired sampling would have been the most robust approach for detecting temporal changes in BD. However, because the 2004 and 2009 datasets were collected from different locations under separate campaigns, a paired t-test was not possible. Instead, we used Welch's t-test (two independent samples with unequal variances), which is more appropriate for independent datasets with unequal sample sizes. We have clarified this point in the revised manuscript by updating Section 2.8 to emphasize why Welch's t-test was selected.

**Ln 287 $\mu_{2009}$ and $\sigma_{2004}$ should be $\sigma_{2009}$ and $\sigma_{2004}$**

**Response:** We have corrected.

**Ln 321: In contrast, the poorest-performing model, PSOC8, exhibited an RMSE of 6.273 g cm⁻³, highlighting significant predictive errors (Fig. 4). The RMSE is far beyond the maximal value of BD of soils. Can it be that wrong units for in- or output variables were used?**

**Response:** we recheck unit again and we found that united correct but PSOC8 had OC and percentage of clay (cl) as predictor in the equation, and this cannot handle very high OC it will predict BD so high such as in inorganic soil type.

**Ln 398: 'This increase may be attributed to factors such as intensified land management practices, reduced soil organic matter, or increased soil compaction over time.' It would be important to discuss how intensified land management practices and soil compaction are related to variables that are used as input in the ANN.**

**Response:** Since Section 3.5 presents results, we kept it unchanged, but we revised Section 4.2 (second paragraph) to clarify the connection between land management practices and ANN input variables. Specifically, we added an explanation of how predictors such as NDVI, BSI, NDMI, slope, and temperature act as proxies for management-driven processes, including vegetation removal, bare soil exposure, soil compaction, and organic matter decomposition. This revision strengthens the link between remote sensing predictors and land management impacts on BD, making it clear how the ANN model captures management-related effects that PTFs relying only on OC cannot represent.

**Ln 400: 'The minimum BD values also showed a substantial increase, rising from 0.12 g cm⁻³ in 2004 to 0.95 g cm⁻³ in 2009´ if not the same sites were visited, a comparison between the extremes is not very informative.**

**Response:** We thank the reviewer for this valuable comment. We acknowledge the limitation that the soil samples from 2004 and 2009 were not collected at the exact same sites. However, both campaigns covered all soil types and were taken from nearby locations, ensuring broad comparability. While the results should be interpreted with caution, they still provide useful guidance for future research and allow us to observe general trends. We have noted this limitation in the conclusion section for clarity.

**Ln 407: This transformation suggests a reduction in the occurrence of extreme BD values and a more balanced distribution of BD by 2009. See my comment above.**

**Response:** We thank the reviewer for this follow-up. As noted in our response to Ln 400, we recognize that comparisons of extreme values should be interpreted with caution since the sampling sites in 2004 and 2009 were not identical. To address this, we have acknowledged this limitation in the conclusion section. We therefore retained the description in Section 3.5 as part of the reported results but clarified its interpretation in conclusion.

**Ln 447 'Our findings show that OC-based PTFs, while exhibiting strong alignment with the RS-ANN model in mean BD values, displayed higher variability and greater prediction uncertainty, particularly in regions with fluctuating organic matter content' Where is that shown?**

**Response:** We have clarified the text in Section 4.1 by explicitly pointing to Figure 7 and Table 7, which demonstrate the higher variability and uncertainty of OC-based PTFs compared to the ANN model.

**Ln 458 where extreme BD values can lead. Do you mean extreme OM?**

**Response:** We have corrected the text to state that the issue arises from extreme OM values leading to unreliable BD estimates, not extreme BD values.

**Ln551: 4) 'Skewness and kurtosis analyses revealed that the RS-ANN model improved from a highly skewed distribution in 2004 (skewness = -2.81, kurtosis = 15.37) to a more balanced distribution in 2009 (skewness = -0.58, kurtosis = -0.41).' If I understand it correctly, this is the skewness of the distribution of predicted BD, and not of the distribution of the difference between observed and predicted BD. It is interesting to note that the ANN and PTFs predict a different distribution**

**Response:** We confirm that the reported skewness and kurtosis values refer to the distribution of predicted BD values, not residuals. To avoid confusion, we revised the text to explicitly state that these are the skewness and kurtosis of predicted BD.

**Ln 553: 'In contrast, PTFs continued to show high skewness and kurtosis, indicating persistent prediction errors for outliers.' This statement is not in line with the results shown in table 7.**

**Response:** We revised the text to more accurately reflect Table 7. The revised sentence now states that PTFs showed higher skewness and kurtosis than ANN in 2004, but that the distributions became more comparable in 2009, avoiding the previous overstatement about persistent prediction errors.

**Ln 555: ´5) The RS-ANN model demonstrated broader applicability across diverse soil types and land uses compared to traditional PTFs and OC-dominant ML models.´ Where is this shown?**

**Response:** We have revised the sentence to explicitly reference the supporting evidence. The broader robustness of the ANN model is demonstrated by the feature importance analysis (Figure 6), which shows its use of multiple predictors (NDVI, NDMI, slope, temperature) compared to the OC-dominant PTFs.

We look forward to hearing from you in due time regarding our submission and to respond to any further questions and comments you may have.

Sincerely,
Sunantha Ousaha
2 October 2025

---

## Author Comment (AC2)

**Author Response to Reviewer 2 Comments**

**Authors:**

We sincerely thank you and the reviewers for the time and effort invested in evaluating our manuscript. We greatly appreciate the constructive comments and insightful suggestions, which have contributed to improving the clarity, accuracy, and overall quality of our work. Below, we provide detailed, point-by-point responses to all major and minor comments.

**Reviewer:**

This manuscript presents a comprehensive and timely study on the estimation of soil bulk density (BD) using machine learning (ML) and remote sensing data, with a specific focus on temporal changes in Thailand between 2004 and 2009. The authors are to be commended for the extensive comparison undertaken, evaluating six different ML models against a very large benchmark of 76 published pedotransfer functions (PTFs). The use of Bayesian Optimization for hyperparameter tuning represents a rigorous and state-of-the-art approach. The paper is well-structured, the research question is significant, and the results, if validated, would be a valuable contribution to the fields of soil science, remote sensing, and land management.

However, there are several major concerns, primarily methodological, that must be addressed before the manuscript can be considered for publication. The most critical issue is a fundamental contradiction between the model development/validation and its application for temporal analysis, which currently undermines the paper's main conclusions regarding temporal trends.

**Author Response:** We acknowledge the concern regarding a potential contradiction between model development/validation and the temporal analysis, and we address this point in detail under the Major Comments below.

**Major Comments:**

**Comment 1:**

The title is misleading because the temporal variation of BD is treated in only one subchapter (Section 3.5), therefore I suggest to modify the title accordingly.

**Author Response:** We agree that the original title may have overemphasized the temporal aspect of the study. We have revised the title to better reflect the broader focus of the manuscript, which includes model benchmarking, feature importance, and temporal transferability. We propose the revised title "Estimating changes in soil bulk density in Thailand using machine learning and remote sensing: model performance, feature importance, and temporal transferability".

**Comment 2:**

The contradiction in the application of the ANN model for 2009 Predictions is the most significant concern. The authors establish that the Artificial Neural Network (ANN) model is superior to other models, including tree-based methods like Random Forest and XGBoost. The feature importance analysis (Section 3.4, Figure 6) is key to this conclusion, showing that the ANN model uses a balanced set of predictors (slope, temperature, vegetation indices, etc.) and does not overly rely on Organic Carbon (OC). This is presented as a major strength, making the model more robust and generalizable. However, when applying this model to the 2009 dataset for temporal analysis, the authors state: "...utilizing only OC data as the sole predictor for BD, as no ground-truth BD measurements were available for validation in that year". This is a critical methodological flaw. The validated high-performance ANN model is a multivariate model that relies on a suite of remote sensing, topographic, and climate inputs. It cannot be applied using only a single input variable (OC). The authors need to clarify precisely how the 2009 predictions were made. Did they train a new, univariate ANN model using only OC? If so, its performance is unknown and unvalidated, and it cannot be claimed to be the "best-performing model.". Did they apply the original multivariate model but feed it only OC data, with placeholder values (e.g., zero, mean) for all other inputs? This would be invalid and produce meaningless results. As it stands, the entire temporal analysis (Section 3.5), including the reported 7.27% increase in mean BD and the 41.23% reduction in standard deviation, is not supported by the methodology. The conclusions about increased soil compaction and reduced variability are therefore unsubstantiated. The authors must either provide the full suite of predictor variables for 2009 and re-run the analysis or retract the temporal claims.

**Author Response:** Our wording created ambiguity about the 2009 prediction. To clarify, the 2009 projections used the full multivariate predictor set, processed identically to 2004 (remote-sensing indices, topography, climate). The 2004-trained ANN (frozen weights) was applied to the 2009 stack; no retraining, placeholders, or constant imputations were used. We have revised Section 2.7 accordingly.

**2.7 Model Implementations and Comparative Analysis**

We trained six machine-learning models on the 2004 dataset, tuned hyperparameters via Bayesian Optimization, and selected the best model using RMSE, MAE, and R², verifying interpretability with permutation feature importance and partial-dependence style summaries to confirm balanced use of predictors. To assess portability across years, we constructed a 2009 multivariate predictor stack using the same variables and preprocessing as in 2004 (static terrain covariates reused; climate summaries and remote-sensing indices recomputed with identical definitions and compositing) and standardized all 2009 predictors with 2004 training statistics to prevent leakage; the frozen 2004-trained ANN was then applied to generate out-of-sample BD projections, produced only where the full predictor set was available. To examine year-specific sensitivity, we report permutation feature importance on the 2009 projections (as model-based sensitivity, not validation). For context, we also estimated 2009 BD using the top-performing PTFs from 2004 where inputs were available and compared distributional statistics (mean, SD, CV) between the PTF estimates and the ANN

projections, noting that 2009 field BD data were unavailable for independent validation.

**Comment 3:**

Equation (1) on page 3 for calculating bulk density. The multiplication by 100 is incorrect. Soil bulk density is a measure of mass per unit volume, with standard units of g cm-3. Multiplying by 100 would make the values physically meaningless (e.g., the reported mean of 1.28 g cm-3 would become 128). This appears to be a significant typo that should be corrected. Please verify if this error propagated into any calculations or if it is merely a display error in the formula.

**Author Response:** We note for completeness that, following Reviewer 1's recommendation, we have removed Equations (1) and (2) from the manuscript. The previously shown "×100" in former Equation (1) was a typesetting error; all computations used correct units (g cm-3) and were unaffected. After removal, the remaining equation (formerly Equation (3)) has been retained and renumbered for consistency. This also addresses the units concern raised in your comment. The corrected definition is:

$$BD = \frac{\text{dry soil mass (g)}}{\text{volume of cylinder (cm}^3)}$$
 (1)

**Minor Comments:**

**Comment 1:** Provide more information on satellite images: how many satellite products did you use in the data analysis after pre-processing? What's the satellite overpass frequency?

**Author Response:** We have added a dedicated paragraph in Section 2.2.1 *Landsat 5 Thematic Mapper (TM) and Pre-processing* detailing sensor characteristics, scene counts, and revisit as follows:

2.2.1 Landsat 5 Thematic Mapper (TM) and Pre-processing
We used Landsat 5 Thematic Mapper (TM) Level-2 surface reflectance imagery for
2004 and 2009 (nominal 16-day revisit; 30 m spatial resolution; 6 spectral bands
in the VIS, NIR, and SWIR). Data were accessed via the Google Earth Engine
catalog. Given the study area's persistent tropical cloud cover, we ingested the full
annual archive (1 January-30 December) and applied standard QA/CFMask-based
cloud-shadow masking and basic BRDF/topographic normalization to ensure
consistent inputs. The annual inventories comprised 769 scenes (2004) and 834
scenes (2009) prior to quality filtering. We then generated annual median
composites (and derived spectral indices used as predictors) to mitigate residual
clouds and temporal noise, yielding stable inputs for model training (2004) and
out-of-sample projection (2009).

Comment 2: In Section 2.8, temporal uncertainty is quantified as the absolute difference in standard deviations between the two years (U= $|\sigma 2009-\sigma 2004|$ ). While this measures the change in the *variability* or *dispersion* of BD predictions, it is not a standard definition of model uncertainty (which typically refers to prediction intervals or confidence in the estimates). The authors should consider rephrasing this to "change in spatial variability" to avoid confusion with predictive uncertainty.

**Author** Agreed. We have rephrased "temporal uncertainty" to "change in spatial variability" throughout Section 2.8, as recommended.

**Comment 3:** The correlation matrix (Figure 2) shows an exceptionally strong negative correlation between OC and BD (r=-0.92). This suggests that OC explains over 84% ( $R^2 \approx 0.85$ ) of the variance in BD in the 2004 dataset by itself. This may limit the generalizability of the findings to regions where this relationship is less dominant. It would be beneficial for the authors to briefly discuss this in the context of their dataset and how it might influence model performance comparisons.

**Author Response:** We agree. We added a brief dataset-level note in Section 3.1 clarifying that the 2004 predictive signal is strongly driven by OC and that the results should be interpreted as a benchmark for that year's covariate pattern. We also expanded Section 4.2 to explain how this dominance can influence model comparisons, tree-based models tend to rely more on OC, whereas the ANN distributes importance more evenly, thus part of the ranking is contingent on the 2004 covariate structure.

**3.1 Descriptive Statistics**

"...The 2004 data exhibit a dominant OC-BD coupling (r = -0.92;  $R^2 \approx 0.85$ ), while all other predictors show weak correlations ( $r \leq 0.19$ ). This pattern indicates that much of the predictive signal in 2004 is attributable to OC within this dataset's covariate structure (climate, topography, and spectral indices for that year). Consequently, the 2004 results should be interpreted as a benchmark specific to this covariate pattern, rather than as evidence that OC will dominate to the same extent in other regions or years."

**4.2 Sensitivity of ML Models to Input Variables and Their Influence on Predicted BD Changes Over Time**

"...Given the very strong OC-BD coupling in the 2004 dataset (Figure 2), part of the in-year accuracy of these OC-dominated models may reflect this dataset-specific relationship rather than a broadly generalizable pattern. In contrast, the ANN model demonstrated a more balanced use of predictors and maintained similar feature-importance behavior when applied to the 2009 dataset (Table 8), suggesting that it can adapt well to interannual variations. This finding indicates that the ANN model's predictive signal is not dominated by a single covariate, supporting its relative stability and potential transferability across different

temporal contexts, although we interpret 2009 results as model-based projections pending ground validation."

**Comment 4:** Fix an objective evaluation for the RMSE obtained from different models. De Vos (2005) established satisfactory prediction performance for RMSE less than 0.25 g cm-3 (Palladino et al., 2022)

**Author Response:** We have added this benchmark in Section 3.2.

Comment 5: Figures: In caption for Fig. 1b, please clarify what the color bar is referring to. I see the red circles in the texture triangle simply indicate the soil samples. Then I see orange circles and colored texture classes. This is confusing. In Fig. 2 the authors should add the color bar title ("correlation"). The caption for Figure 4 reads "Loss function curves for neural network regression models... and learning curves for other machine learning models...". However, Figure 4 only shows these curves. The scatterplots are in Figure 5. The caption for Figure 4 should be corrected to only describe its own content. Increase font size of the axis tick labels, add a grid. In Fig. 5 increase font size of the axis tick labels, add a grid

**Author Response:** OK, we will fix this in the revised version.

**Comment 6:** Tables: In Table 2 replace "TPFs" with "PTFs". In Table 3, please add the coefficient of variation (CV). In Table 6, please specify how many data were used for calibration (training) and how many for validation (testing)

**Author Response:** Yes, we edited in Table 2, added coefficient of variation (CV) in Table 3, and added detail about training (189 soil samples) and testing (47 soil samples) dataset in Table 6.

**Comment 7:** Inconsistent Units in Text: The manuscript occasionally uses inconsistent formatting for units. For example, in the abstract, the RMSE is given as "0.017 g cm³", while in the ML performance table (Table 6), the MAE for the ANN model is listed as "0.012" without clear units in the text, and elsewhere as "0.012 g cm→". Please ensure consistent formatting (g cm⁻³) throughout the manuscript for clarity and professionalism.

**Author Response:** Thank you, we will fix this in the revised version.

**Recommendation:**

**I recommend Major Revisions.**

The manuscript addresses an important research topic and employs a robust and extensive comparative framework. The rigorous hyperparameter tuning and the scale of the PTF benchmark are significant strengths. However, the foundational contradiction in the

methodology for the 2009 temporal analysis is a critical flaw that invalidates the paper's primary conclusions regarding temporal trends in soil bulk density.

The authors must resolve this issue by either providing a valid methodology for the 2009 predictions using the full multivariate ANN model or by reframing the paper to focus solely on the 2004 model comparison, removing the temporal analysis. Additionally, the error in the BD formula and other minor points should be addressed. If the authors can satisfactorily resolve these major concerns, the revised manuscript would likely my opinion be suitable for publication.

**REFERENCES**

De Vos, B., Van Meirvenne, M., Quataert, P., Deckers, J., Muys, B., 2005. Predictive quality of pedotransfer functions for estimating bulk density of forest soils. Soil Sci. Soc. Am. J. 69 (2), 500–510.

Palladino M., N. Romano, E. Pasolli, P. Nasta. 2022. Developing pedotransfer functions for predicting soil bulk density in Campania Region. Geoderma 412, 115726 https://doi.org/10.1016/j.geoderma.2022.115726

Citation: https://doi.org/10.5194/egusphere-2025-2360-RC2

**Author Response:** We would like to clarify that our analysis for 2009 was conducted using the same full multivariate ANN model developed for 2004, incorporating all predictor variables (remote sensing indices, topographic, climatic, and soil covariates). However, the predictor datasets were recomputed using the 2009 environmental and remote sensing conditions, rather than reusing 2004 data. We have addressed this point explicitly in the Major Comments section for clarity. In addition, we have carefully addressed all minor comments to improve the manuscript's clarity and scientific rigor.

We hope that we have been able to satisfactorily address the issues that have been raised above from the reviewers. We look forward to hearing from you.

Sincerely, Sunantha Ousaha On behalf of all co-authors 10 October 2025